# Unexpected sequences and structures of mtDNA required for efficient transcription from the first heavy-strand promoter

Akira Uchida[1†], Divakaran Murugesapillai[2†], Markus Kastner[1], Yao Wang[1], Maria F Lodeiro[1], Shaan Prabhakar[1], Guinevere V Oliver[1], Jamie J Arnold[1], L James Maher III[3], Mark C Williams[2*], Craig E Cameron[1*]

[1]Department of Biochemistry and Molecular Biology, The Pennsylvania State University, State College, United States; [2]Department of Physics, Northeastern University, Boston, United States; [3]Department of Biochemistry and Molecular Biology, Mayo Clinic College of Medicine, Rochester, United States

**Abstract** Human mtDNA contains three promoters, suggesting a need for differential expression of the mitochondrial genome. Studies of mitochondrial transcription have used a reductionist approach, perhaps masking differential regulation. Here we evaluate transcription from light-strand (LSP) and heavy-strand (HSP1) promoters using templates that mimic their natural context. These studies reveal sequences upstream, hypervariable in the human population (HVR3), and downstream of the HSP1 transcription start site required for maximal yield. The carboxy-terminal tail of TFAM is essential for activation of HSP1 but not LSP. Images of the template obtained by atomic force microscopy show that TFAM creates loops in a discrete region, the formation of which correlates with activation of HSP1; looping is lost in tail-deleted TFAM. Identification of HVR3 as a transcriptional regulatory element may contribute to between-individual variability in mitochondrial gene expression. The unique requirement of HSP1 for the TFAM tail may enable its regulation by post-translational modifications.

*For correspondence: ma. williams@northeastern.edu (MCW); cec9@psu.edu (CEC)

†These authors contributed equally to this work

Competing interests: The authors declare that no competing interests exist.

## Introduction

In spite of the absolute requirement of mitochondrial function for life, more is known about gene-regulatory mechanisms in prokaryotes than is known about corresponding mechanisms in the mitochondria of humans or other mammalian species. One reason for this knowledge gap is the inability to perform reverse-genetic analysis of mitochondrial DNA (mtDNA). Studies performed in cells and cell-free extracts have revealed the existence of three mitochondrial promoters: light-strand promoter (LSP), heavy-strand promoter 1 (HSP1) and HSP2 (*Bogenhagen et al., 1984*; *Cantatore and Attardi, 1980*; *Chang and Clayton, 1984*; *Montoya et al., 1982*, *1983*). Transcripts from each promoter are polygenic and all contain tRNAs (*Ojala et al., 1980*, *1981*). mtDNA encodes only 13 proteins, all components of the electron transport chain (ETC) or ATP synthase. The mRNA for ND6 is transcribed from LSP; the other 12 mRNAs are transcribed from HSP2 (*Clayton, 1984*). The primary role of HSP1 is transcription of rRNA genes (*Clayton, 1984*). We recently suggested that such a division of transcription would allow assembly of the core components of the ETC by activating HSP2. In this model, activation of complex I would not occur until LSP was activated, as ND6 mRNA is the sole mRNA transcribed from this promoter. Such an order could diminish reactive oxygen production that might be produced if ND6 subcomplexes formed (*Lodeiro et al., 2012*). Having rRNA

transcription controlled distinctly by HSP1 would permit mitochondrial biogenesis to be regulated distinctly from the myriad other homeostatic functions of mitochondria (*Gaines and Attardi, 1984*; *Gaines et al., 1987*; *Lodeiro et al., 2012*).

For a regulated program of gene expression to exist, each promoter should exhibit some unique attribute relative to the others with respect to transcription (re)initiation and/or elongation, steps most frequently used to control transcription. The first promoter characterized biochemically was LSP. This initial characterization showed that the region 50 bp upstream of the transcription start site was sufficient for transcription (*Bogenhagen et al., 1984*; *Fisher and Clayton, 1985*; *Fisher et al., 1987*; *Gaines and Attardi, 1984*). In going from cell-free extracts to highly purified systems, the genetic determinants for LSP did not change (*Falkenberg et al., 2002*; *Gaspari et al., 2004*; *Lodeiro et al., 2010*; *Sologub et al., 2009*). When the genetic architecture of HSP1 was deduced based on LSP, factor-dependent transcription could be observed (*Bogenhagen et al., 1984*). The factors required for initiation are mitochondrial transcription factor A (TFAM), mitochondrial transcription factor B2 (TFB2M), and the mitochondrial RNA polymerase (POLRMT) (*Falkenberg et al., 2002*; *Fisher and Clayton, 1985*; *Fisher et al., 1987*). The model that has emerged for transcription initiation from LSP posits binding of one molecule of TFAM at a specific binding site upstream of the transcription start site (*Gaspari et al., 2004*; *Lodeiro et al., 2010*; *Morozov et al., 2015*; *Sologub et al., 2009*). This binding event leads to a large bend, 'U-turn,' of the DNA (*Ngo et al., 2011*; *Rubio-Cosials et al., 2011*). POLRMT is then recruited to the promoter by means of an interaction of its first 150 amino acids with TFAM (*Morozov et al., 2014*). TFB2M likely joins with POLRMT as these proteins form a stable complex (*Gaspari et al., 2004*). Worth noting, binding of TFB2M to POLRMT is not essential for its recruitment by TFAM (*Morozov et al., 2014*).

There is a consensus for this mechanism of transcription initiation from LSP (*Gaspari et al., 2004*; *Lodeiro et al., 2010*; *Morozov et al., 2015*; *Sologub et al., 2009*). However, how initiation from HSP1 and HSP2 occurs is actively debated (*Litonin et al., 2010*; *Lodeiro et al., 2012*; *Morozov and Temiakov, 2016*; *Shi et al., 2012*; *Shutt et al., 2010*; *Zollo et al., 2012*). Some investigators observe transcription only from HSP1 in the presence of all three factors, and observe no transcription from HSP2 (*Litonin et al., 2010*; *Morozov and Temiakov, 2016*; *Shi et al., 2012*). Other investigators observe transcription from HSP1 and HSP2 in the presence of only POLRMT and TFB2M (*Lodeiro et al., 2012*; *Shutt et al., 2010*; *Zollo et al., 2012*). Interestingly, these same investigators show that TFAM stimulates transcription from HSP1 but inhibits transcription from HSP2 (*Lodeiro et al., 2012*; *Zollo et al., 2012*). Therefore, some conclude that there is no regulation of transcription mediated by the core components of transcription (*Litonin et al., 2010*; *Morozov and Temiakov, 2016*; *Shi et al., 2012*). Others conclude that POLRMT and TFB2M represent the core transcription machinery that is regulated by TFAM (*Lodeiro et al., 2012*; *Shutt et al., 2010*; *Zollo et al., 2012*). In this latter scenario, TFAM is essential for any transcription from LSP, an activator of transcription from HSP1 and an inhibitor of transcription from HSP2 (*Lodeiro et al., 2012*).

Essentially all of studies of mitochondrial transcription using purified proteins rather than cell-free extracts rely on minimal promoters in isolation. The objective of this study was to evaluate the mechanism of transcription of LSP and HSP1 simultaneously using a dual-promoter template in the same context in which these promoters appear in mtDNA. Data obtained with this dual-promoter template show unambiguously that LSP and HSP1 exhibit unique properties that permit each to be regulated independent of the other. Even though TFAM activates both promoters in this context, the concentration dependence and TFAM domain-dependence for this activation differ between the two promoters. In the absence of TFAM, only a single round of transcription occurs from HSP1, suggesting that TFAM activates HSP1 by promoting reinitiation. Unexpectedly, we identified sequences of mtDNA upstream and downstream of the transcription start site for HSP1 that contribute substantially to both basal and activated transcription from this promoter. The sequences upstream of the transcription start site are a part of a region of mtDNA that is hypervariable in the human population (HVR3), the variability of which may predispose to disease (http://www.mitomap.org/MITOMAP). Using atomic force microscopy, we show that TFAM bound to these elements of HSP1 multimerize in a manner dependent on the carboxy-terminal tail of TFAM. Formation of the resulting DNA loops correlated directly to maximal activation of transcription from HSP1. Similarly, template compaction observed at higher TFAM concentrations correlates with transcription inhibition (*Farge et al., 2014*). Our data point to the potential for post-translational modifications of the TFAM carboxy-terminal

tail to regulate mitochondrial gene expression. We further suggest that HVR3 polymorphisms may contribute to disease by interfering with regulated expression of the mitochondrial genome.

## Results

The light-strand promoter (LSP) is the most extensively studied human mtDNA promoter. All of the cis-acting elements required for transcription factor-dependent initiation are located within 50 bp upstream of the transcription start site. The primary role of this upstream sequence is to bind mitochondrial transcription factor A (TFAM), a DNA binding, bending and wrapping protein related to mammalian high-mobility-group proteins (*Malarkey and Churchill, 2012*; *Murugesapillai et al., 2017*). It has been assumed by some that the organization and TFAM-dependence of the HSP1 and HSP2 promoters are identical. Our previous studies, however, suggested that TFAM regulation of HSP1 and HSP2 differs (*Lodeiro et al., 2012*).

Sequences more than 50 bp upstream of the transcription start sites of LSP and HSP1 have largely been ignored in studies of mitochondrial transcription, at least over the past decade. This inter-promoter region (IPR) is the third of three hypervariable regions (HVR3) of mtDNA (*Figure 1a*). We reasoned that direct comparison of the two promoters would be facilitated by studying a relevant mtDNA fragment. We chose a sequence that began 35 bp downstream of LSP, producing a 35-nt transcription product, and ended 45 bp downstream of HSP1, producing a 45-nt transcription product (*Figure 1b*). We refer to this DNA fragment as the dual-promoter template. Transcription using the dual-promoter template employed a $^{32}$P-labeled adenylate homotrimer as primer in order to quantify transcription products after separation by denaturing polyacrylamide gel electrophoresis (*Figure 1—figure supplement 1*). The adenylate homotrimer, pAAA, allows for efficient transcription initiation, whereas the use of a homodimer, pAA, results in either slippage synthesis or the production of a dead-end product, pAAG (*Figure 1—figure supplement 2*). Reaction conditions used were stringent and consistent with those used to study a variety of nucleic acid polymerases (*Figure 1—figure supplement 1*). Under these conditions, transcription products formed linearly for at least 30 min (*Figure 1—figure supplement 1*). Most single time-point assays were performed between 5 and 15 min.

The first experiment used the dual-promoter template to evaluate transcription from LSP and HSP1 as a function of TFAM concentration. Transcription from HSP1 was readily detectable in the absence of TFAM (*Figure 1c*); the yield ranged from 70 to 100 nM, which is essentially stoichiometric with template (*Figure 1d*). The TFAM-independent yield from HSP1 did not change even after a 90 min incubation (*Figure 1—figure supplement 3*). Titration of TFAM into the reaction resulted in an increase in transcription from LSP that reached a maximum value of ~500 nM transcript at a TFAM concentration of 300 nM (*Figure 1c and d*). Over this same range, output from HSP1 changed minimally. Transcription from HSP1 reached a maximum value of ~300 nM transcript at a TFAM concentration of 1000 nM (*Figure 1c and d*). For TFAM concentrations at which transcription from LSP was inhibited almost entirely, transcription from HSP1 proceeded to levels no less than 50% of maximum levels (*Figure 1c and d*). Together, these observations demonstrate different requirements for maximal transcription from LSP and HSP1. Some of these differences are due to differential requirements for and response to TFAM.

In order to determine the extent to which the IPR contributed to LSP and HSP1 function, we randomized the IPR sequence. Interestingly, randomization exhibited a much more pronounced effect on transcription from HSP1 than from LSP (*Figure 1—figure supplement 1f*). These data suggest that the IPR is a functional component of the HSP1 core promoter. In order to further delimit the sequences of the IPR that contribute to HSP1 and/or LSP function, we constructed single-promoter templates that included the IPR through the distal TFAM-binding site (*Figure 2a*). For HSP1, 138 bp upstream of the transcription start site were present; for LSP, 146 bp upstream of the transcription start site were present (*Figure 2a*). The LSP TFAM-binding site (region one in *Figure 2a*) could be deleted without any effect on transcription from HSP1, maintaining the 4-fold activation by TFAM observed for the dual-promoter template (−114 in *Figure 2b*). However, additional deletions interfered with TFAM-activated transcription from HSP1 (−60 and −50 in *Figure 2b*). Extending LSP by adding sequences beyond −50, exhibited an ~50% to 75% increase in TFAM activation (−69,–86, −115 in *Figure 2c*). When the deletion construct was extended to include region 5 (-146 in *Figure 2c*) there was an ~30% decrease in transcription from LSP compared to those constructs that

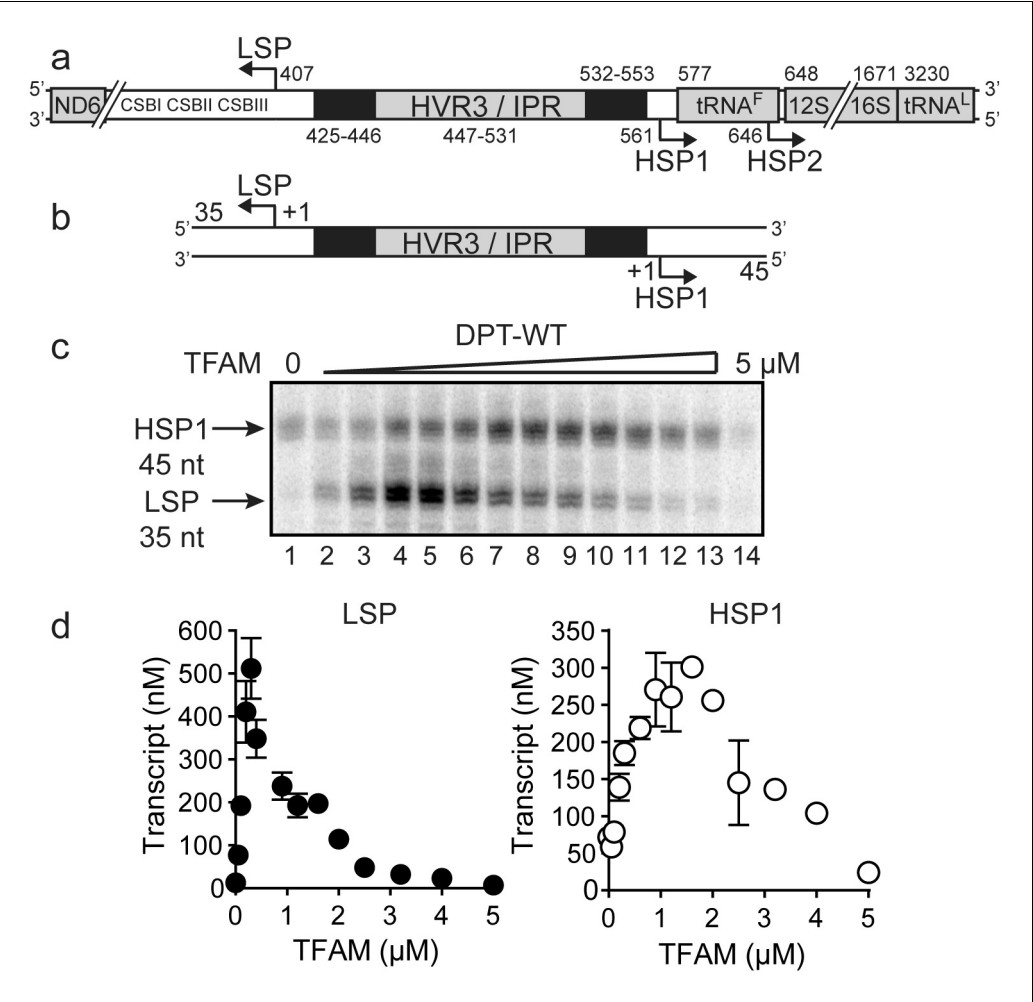

**Figure 1.** Dual promoter template. (**a**) Schematic of the mtDNA transcriptional control region including the three promoters: LSP, HSP1 and HSP2; and interpromoter region (IPR), also known as hypervariable region 3 (HVR3). The relative positions of conserved sequence boxes (CSB I, II and III), ND6, tRNA[F], 12S and 16S rRNA genes are shown. Numbering according to the standard Cambridge mtDNA sequence. Black boxes indicate the putative TFAM binding sites. (**b**) Dual promoter DNA oligonucleotide (234 bp) template containing LSP, HSP1 and HVR3/IPR used for in vitro transcription reactions. This oligo gives rise to LSP and HSP1 derived RNA transcripts 35 and 45 nts long, respectively. (**c**) Run-off transcription products using the dual promoter DNA oligo template (DPT-WT) and increasing concentrations of TFAM (0–5 μM) resolved by denaturing PAGE. (**d**) Amount of LSP (35 nt) and HSP1 (45 nt) RNA transcripts produced using the dual promoter construct plotted as a function of TFAM concentration. Data are means from three independent experiments. *Error bars* represent ± S.E.M.

The following figure supplements are available for figure 1:

**Figure supplement 1.** Dual promoter template: Time course and reaction conditions.

**Figure supplement 2.** The adenylate homotrimer, pAAA, is specifically and efficiently used for transcription initiation, whereas the pAA and pAAG primers are not.

**Figure supplement 3.** Dual promoter construct: TFAM-independent activity.

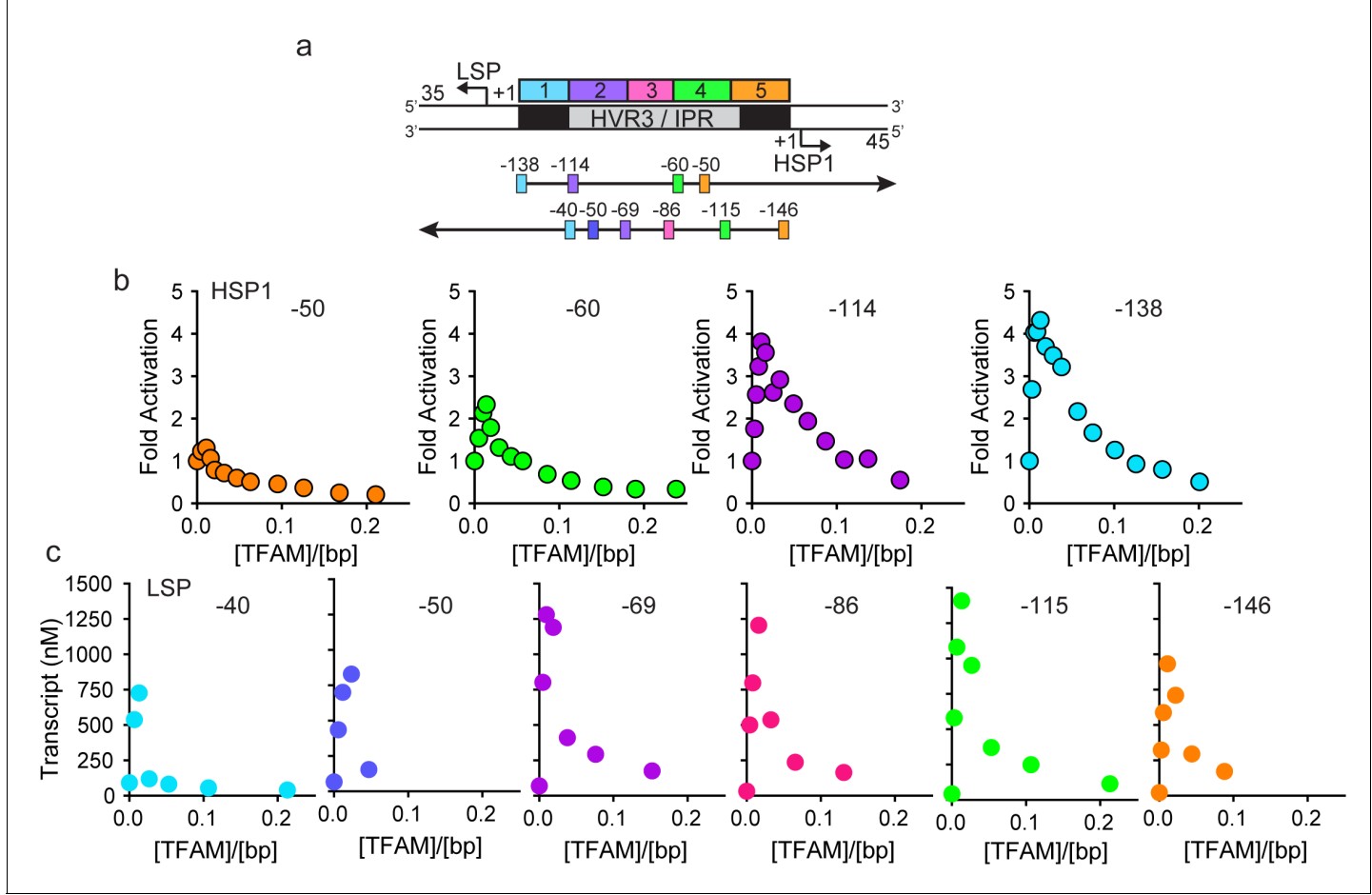

**Figure 2.** HVR3/IPR contributes to transcription from HSP1. (a) Deletion constructs used to assess the impact of HVR3/IPR on LSP and HSP1 transcription. The section between and including the putative TFAM binding sites was divided into five distinct regions (1-5; 1:425–447; 2:448–476; 3:447–493; 4:494–522; 5:523–550) and used to guide the deletion design strategy. Four (−50,−60, −114 and −138) and six (−40,−50, −69,−86, −115 and −146) different deletion constructs were used to assess HSP1 and LSP transcription, respectively. (b) Fold activation on HSP1 transcription plotted as a function of TFAM concentration per bp for the different HSP1 deletion constructs. (c) Amount of LSP transcript product plotted as a function of TFAM concentration per bp for the different LSP deletion constructs.

did not include this region (−69,−86 and −115), suggesting that region five has a modest negative effect on the output from LSP when the IPR sequence is present. We conclude that the sequence extending to 114 bp upstream of the HSP1 transcription start site should be considered an integral component of the TFAM-responsive element for this promoter. For LSP, the long-known TFAM-binding site appears sufficient for transactivation.

As discussed above, activation of LSP by TFAM occurs by binding to the TFAM-binding site (region one in *Figure 2a*), bending the DNA, and recruiting POLRMT and TFB2M (*Gaspari et al., 2004*; *Lodeiro et al., 2010*; *Morozov et al., 2015*; *Rubio-Cosials et al., 2011*). Transcription from LSP is strictly dependent on TFAM. The longstanding paradigm for TFAM-activated transcription from HSP1 suggested a comparable mechanism, with the TFAM-binding site being located between positions 532 and 553 of mtDNA (region five in *Figure 2a*). However, here we show that transcription from HSP1 can occur independent of TFAM, and that the TFAM-responsive element includes more than the proposed HSP1 TFAM-binding site (region five in *Figure 2a*). We therefore addressed the role of regions 1 and 5 using the dual-promoter template. Deletion of region one had no impact on transcription from HSP1 or its activation by TFAM, but essentially silenced transcription from LSP (*Figure 3a*). Similarly, deletion of region five had no impact on transcription from LSP (*Figure 3b*). Interestingly, region five was not required for TFAM-independent transcription from HSP1

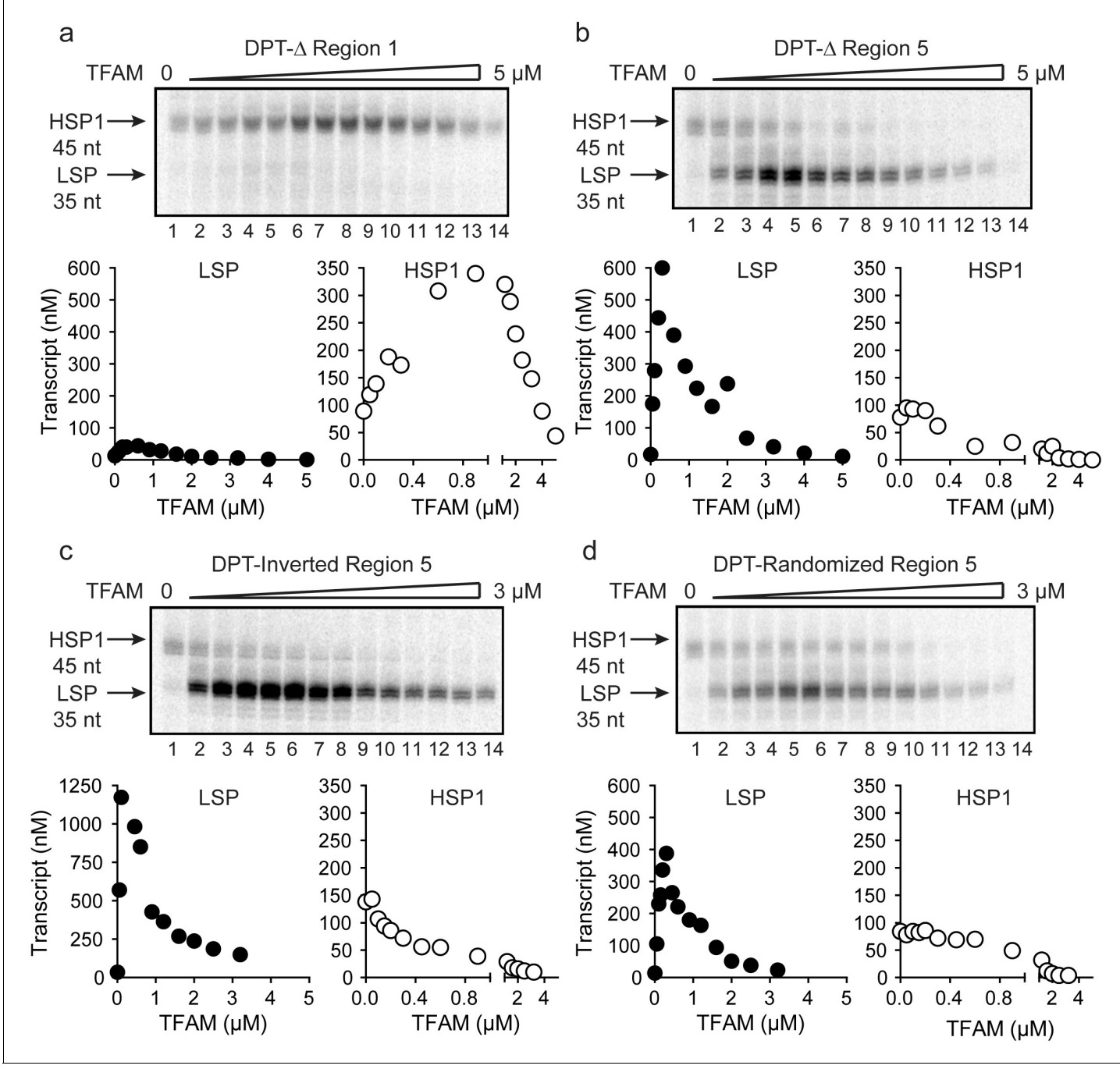

**Figure 3.** TFAM-binding site of record for HSP1 may not actually bind TFAM to contribute to HSP1 transcription. (a) Deletion of LSP TFAM-binding site, Δ Region 1, from dual promoter template precludes transcription from LSP. (b) Deletion of HSP1 TFAM-binding site, Δ Region 5, from dual promoter template does not interfere with TFAM-independent activity but converts TFAM from an activator to an inhibitor. (c,d) Inversion and randomization of region five also converts TFAM into an inhibitor. Shown are the run-off transcription products of LSP and HSP1 promoter-dependent transcription resolved by denaturing PAGE and the amount of transcription product plotted as a function of TFAM for the Δ Region 1, Δ Region 5, Inverted Region five and Randomized Region five constructs.

(*Figure 3b*). Surprisingly, deletion of region five converted TFAM from an activator of HSP transcription to a repressor (*Figure 3b*). One possible explanation is that the deletion created a negative element by sequence juxtaposition. To test this possibility, we inverted region five or randomized the

sequence of region 5. Regardless of the change, TFAM continued to repress transcription from HSP1 instead of activate transcription (*Figure 3c and d*). Worth noting, inverting region five caused a significant increase in the yield from LSP (*Figure 3c*); the basis for this activation requires further study. Our general conclusion from this line of investigation is that region five may be a transcription 'insulator,' a sequence that prevents TFAM bound upstream of region five from influencing transcription downstream where POLRMT-TFB2M assembles.

Historically, region 5 was first hypothesized to be a TFAM binding site based on its location at a position equivalent to that of the LSP TFAM-binding site, similarity to the sequence of the LSP TFAM binding site, and the finding that deletion of the sequence interfered with transcription from HSP1 in cell-free extracts (*Bogenhagen et al., 1984*). Later, more direct evidence for TFAM binding was obtained by DNAse I footprinting (*Fisher et al., 1987*). In order to determine when and where TFAM binds to the dual-promoter template under transcription conditions, and as a function of TFAM concentration, we performed a DNAse I footprinting experiment (*Figure 4*). When the HSP1 template strand was labeled, protection was observed in region 1 as expected. A DNAse I hypersensitive site appears to be due to TFAM bending of DNA at this site (*Figure 4a* and *Figure 4—figure supplement 1a*). Another hypersensitive site was observed at the edge of region 2 (*Figure 4a* and *Figure 4—figure supplement 1a*). There was clear nuclease protection between regions 2–4 (*Figure 4a* and *Figure 4—figure supplement 1a*). Protection in regions 2–4 was observed at lower TFAM concentrations than for region 5 (*Figure 4a* and *Figure 4—figure supplement 1a*). Indeed, even at the highest concentration of TFAM employed, complete protection of region 5 was not observed (*Figure 4a* and *Figure 4—figure supplement 1a*). When the LSP template strand was labeled, similar conclusions were reached (*Figure 4b* and *Figure 4—figure supplement 1b*). In addition to protection in regions 2–4, many sites of hypersensitivity were noted (*Figure 4b* and *Figure 4—figure supplement 1b*). Changes in region 5 were observed at the highest TFAM concentrations (*Figure 4b* and *Figure 4—figure supplement 1b*). These changes included weak protection and the presence of a hypersensitive site (*Figure 4b* and *Figure 4—figure supplement 1b*). These data are consistent with binding of TFAM to regions 2–4 correlating to activation of transcription from HSP1 and binding of TFAM to region 5 correlating to transcription inhibition.

Biophysical and structural studies of TFAM have shown that its carboxy-terminal tail, defined here as the final 26 amino acid residues (221–246, *Figure 5a*), is not only able to interact with DNA (*Figure 5b*) but also able to contribute to interactions with a second molecule of TFAM to form a dimer (*Figure 5c*) (*Wong et al., 2009*). Such intermolecular dimerization of separate DNA-bound TFAM molecules could create DNA loops. Looping could impose DNA strain within the loop and perhaps even induce the DNase I hypersensitivity observed in regions 2–4 (*Figure 4b* and *Figure 4—figure supplement 1b*). TFAM deleted for its carboxy-terminal tail remains competent to bind DNA (*Wong et al., 2009*). We evaluated the ability of tail-deleted TFAM (CTΔ26) to stimulate transcription using the dual-promoter template. This derivative activated transcription from LSP with the same concentration dependence as observed for the wild-type (WT) protein (compare LSP in *Figure 5d and e* to *Figure 1c and d*). Activation of this promoter was, however, reduced by approximately three-fold (compare LSP in *Figure 5e* to *Figure 1d*). In contrast, TFAM-CTΔ26 was completely unable to activate transcription from HSP1 (compare HSP1 in *Figure 5d and e* to *Figure 1c and d*). The transcription-repression activity of TFAM observed at higher ratios of TFAM: dual-promoter template was unchanged for both promoters by deleting the TFAM carboxy-terminal tail (compare *Figure 5d and e* to *Figure 1c and d*). We also evaluated the interaction of TFAM-CTΔ26 with the dual-promoter template by using DNAse I footprinting (*Figure 5—figure supplement 1*). TFAM carboxy-terminal tail truncation eliminated all of the DNAse I-protected and hypersensitive sites in regions 2–4 (*Figure 5—figure supplement 1*). Protection of the LSP TFAM-binding site in region one was not eliminated but was reduced several fold (*Figure 5—figure supplement 1*). Together, these data are consistent with a model in which the TFAM carboxy-terminal tail enables a specific interaction of TFAM with regions 2–4 that induces a specific conformation of regions 2–4, perhaps a DNA loop, that is absolutely essential for TFAM activation of HSP1.

Given the requirement of the TFAM carboxy-terminal tail for activation of HSP1, we were curious to know if the TFAM carboxy-terminal tail contributed to the repression of HSP1 observed when region 5 of the dual-promoter template was altered (*Figure 3*). We performed a transcription reaction using the dual-promoter template containing an inversion of the region-5 sequence (*Figure 5f and g*). Interestingly, by using TFAM-CTΔ26 instead of TFAM, the inhibitory nature of the region-5

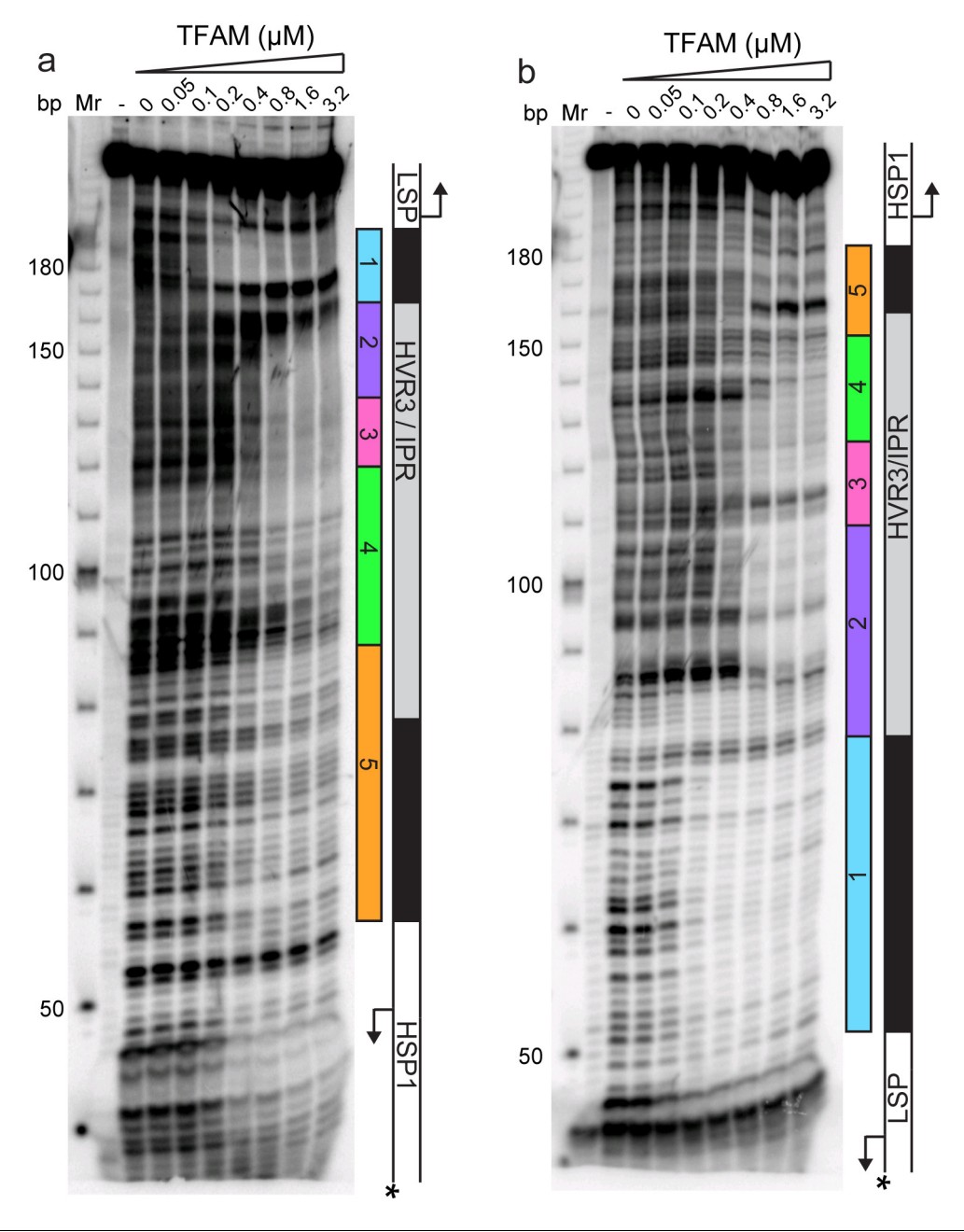

**Figure 4.** Footprinting of the dual promoter template in the presence of TFAM confirms protection or sensitivity in regions 2 and 4 before region 5. DNAse I footprinting of the dual promoter template with increasing concentrations of TFAM. (**a**) HSP1 template strand [32]P-labeled. (**b**) LSP template strand [32]P-labeled. A schematic of the transcriptional control region is shown to the right of the denaturing PAGE gels to indicate regions of protection.

The following figure supplement is available for figure 4:

**Figure supplement 1.** Footprinting of the dual promoter template in the presence of TFAM confirms protection or sensitivity in regions 2 and 4 before region 5.

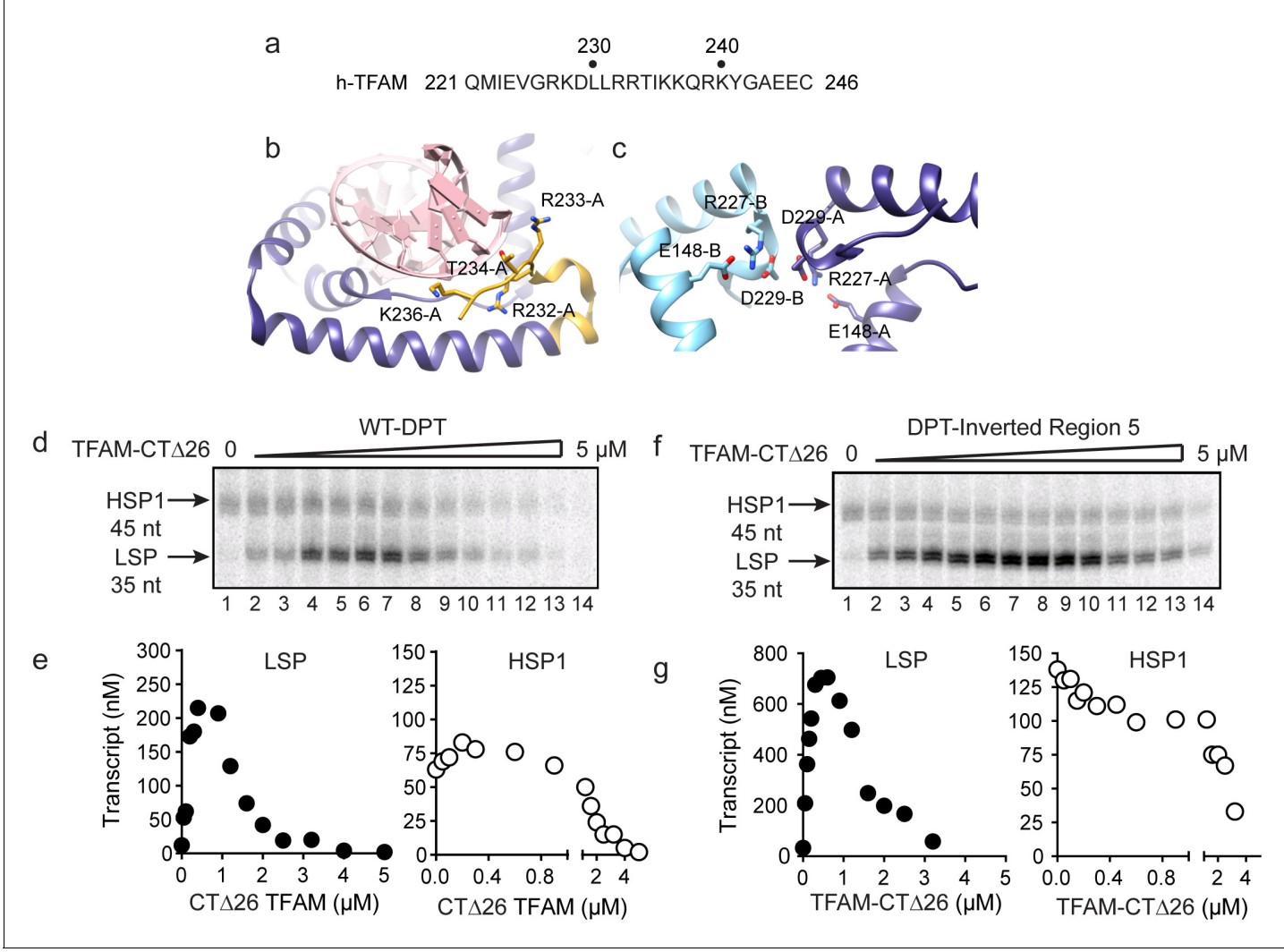

**Figure 5.** Carboxy-terminal tail of TFAM is essential for transcription from HSP1. (**a**) Carboxy-terminal tail (CTT) primary sequence of human TFAM. The last 26 amino acid residues are shown. (**b,c**) Interactions of TFAM CTT. Structural models were produced using PDB 3TQ6 (*Rubio-Cosials et al., 2011*). In panel b, TFAM residues 221–236 are colored yellow. Residues 237 to 246 are disordered and absent in the structure. All other residues of TFAM are colored purple. Residues 232–236 interact with the phosphodiester backbone of bound DNA (red). In panel c, two TFAM-DNA complexes are present in the asymmetric unit and designated here as chain A (dark blue) and chain B (light blue). Structural integrity of the CTT of each monomer benefits from interaction of Arg-227 in each monomer with both Asp-229 and Glu148 of the same monomer. The CTT of one monomer packs against that of a second, perhaps creating a mechanism for association between TFAM-DNA complexes. (**d**) Run-off transcription products using the dual promoter DNA oligo template and increasing concentrations of TFAM-CTΔ26 (0–5 μM) resolved by denaturing PAGE. (**e**) Amount of LSP (35 nt) and HSP1 (45 nt) RNA transcripts produced using the dual promoter construct plotted as a function of TFAM-CTΔ26 concentration. (**f**) Run-off transcription products using the dual promoter DNA oligo template with region five inverted and increasing concentrations of TFAM-CTΔ26 (0–5 μM) resolved by denaturing PAGE. (**g**) Amount of LSP (35 nt) and HSP1 (45 nt) RNA transcripts produced using the dual promoter construct with region five inverted plotted as a function of TFAM-CTΔ26 concentration.

The following figure supplements are available for figure 5:

**Figure supplement 1.** Footprinting of the dual promoter template in the presence of either WT and TFAM-CTΔ26 with POLRMT and TFB2M.

**Figure supplement 2.** Footprinting of the dual promoter template with Inverted Region five in the presence of WT and TFAM-CTΔ26 with POLRMT and TFB2M.

inversion was eliminated (*Figure 5f and g*). Unfortunately, DNAse I footprinting failed to contribute additional insight to our understanding of the mechanism of repression or anti-repression (*Figure 5—figure supplement 2*).

The preceding studies suggest that sequences upstream of HSP1 contribute more to the regulation of this promoter than previously recognized. For completeness, we asked if sequences downstream of the HSP1 transcription start site contributed to promoter regulation. The template used for this experiment contained only regions 2–5 upstream of the transcription start site. The downstream sequence varied from 45 bp, the length used in the dual-promoter template, to 145 bp (*Figure 6a*). Unlike most prior experiments of this type (*Litonin et al., 2010*; *Morozov and Temiakov, 2016*; *Shi et al., 2012*), a $^{32}$P-labeled primer was used instead of an $\alpha$-$^{32}$P-labeled ribonucleotide. Therefore, in our experiments, an increase in labeled product signifies an increased transcript yield rather than more incorporation of labeled ribonucleotide substrate. A clear increase in yield was visible in templates as long as 95 bp; thereafter, the change was more subtle (*Figure 6b*). Yield was dependent on the TFAM concentration (*Figure 6b*). A quantitative perspective is provided in *Figure 6c*. The optimal ratio of TFAM:template for maximal activation was 0.020–0.030 for all templates employed. By increasing the template to include the entirety of the tRNA$^F$ gene, the TFAM-

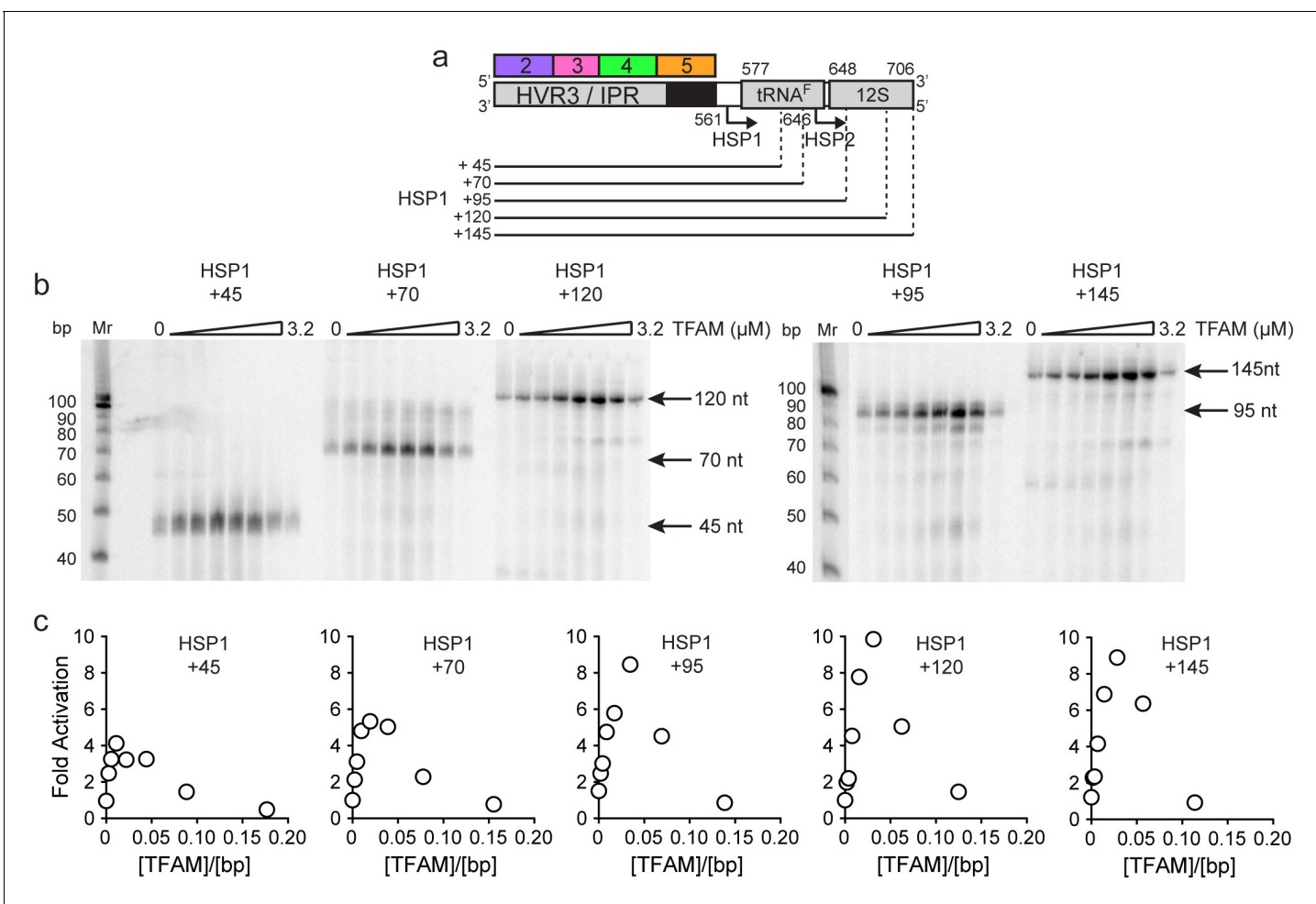

**Figure 6.** mtDNA sequences downstream of HSP1 increase the fold activation on HSP1 transcription. (a) Schematic of HSP1 dsDNA oligonucleotide templates used for in vitro transcription reactions which contained additional mtDNA sequences, either 45, 79, 95, 120 or 145 bp, downstream of the HSP1 promoter start site. Each template contained the HVR3/IPR, starting from Region 2, and ending at the indicated bp from the HSP1 start site. The dotted lines show the approximate position each DNA oligo template end relative to the tRNA$^F$ and 12S RNA genes. (b) Run-off transcription products using HSP1 DNA oligo templates and increasing concentrations of TFAM (0–5 µM) resolved by denaturing PAGE. Ten-bp markers are indicated on each gel. (c) Fold activation on HSP1 transcription plotted as a function of TFAM concentration using the different HSP1 DNA oligo templates.

dependent yield from HSP1 was more than 8-fold higher than the TFAM-independent yield (*Figure 6c*). We conclude that sequences downstream of the HSP1 transcription start site contribute to the output of this promoter, and perhaps also its regulation. Because the TFAM-independent yield remained stoichiometric with template for all constructs tested, downstream sequence may increase RNA yield by somehow promoting template reutilization.

Our data are consistent with a model in which mtDNA sequences from 447 to 656 (see *Figure 1a*) contribute to HSP1 transcriptional output in the presence of TFAM. This stretch of mtDNA begins with HVR3, includes the tRNA$^F$ gene and ends near the 5' terminus of the 12S rRNA gene. To our knowledge, HVR3 has never been implicated in any aspect of mitochondrial gene expression. Because the same carboxy-terminal 26 amino acid residues of TFAM required for HVR3-dependent transactivation of HSP1 (*Figure 5*) are required for dimerization and DNA looping (*Ngo et al., 2011*; *Rubio-Cosials et al., 2011*), it was possible that TFAM-mediated looping in HVR3 contributed to HSP1 transactivation. We used atomic force microscopy to investigate the complex formed upon addition of TFAM or TFAM-CTΔ26 to a 1653 bp DNA fragment, comprising the dual-promoter template (153 bp) extended upstream of LSP by 500 bp and downstream of HSP1 by 1000 bp. We will refer to this fragment here as mtDNA. An irrelevant sequence of similar length (1650 bp) was derived from pUC18 and used as a control for specificity. This DNA is termed pUC. We imaged samples of mtDNA in the absence (*Figure 7—figure supplement 1*) and presence (*Figure 7*) of TFAM using an APTES-coated mica surface and imaging in liquid. In this method, the DNA is kinetically trapped in a configuration that reflects its three dimensional solution conformation (*Murugesapillai et al., 2017a*). In the absence of TFAM, we measured an average contour length of 1684 ± 21 bp, consistent with the sequence used. We titrated mtDNA with TFAM, from 0.05 to 2.7 molecules of TFAM per bp. The lowest ratio is near optimal for transcription from HSP1; the highest ratio is inhibitory (*Figure 6c*). At 0.05 molecules of TFAM per bp, loops were clearly visible (*Figure 7a*). Further addition of TFAM converted loops into more compact structures (*Figure 7a*). We used effective contour length to monitor this process quantitatively. A 4-fold reduction in effective contour length was observed in the presence of the highest concentration of TFAM (*Figure 7b*).

The observation of loops at TFAM concentrations that promote maximal transcription is provocative. We then sought to determine the location(s) of the loops. We repeated the experiment at the lowest ratio of TFAM: mtDNA to obtain sufficient looped molecules for detailed characterization and statistical analysis (*Figure 7—figure supplement 2a,b*). For each mtDNA observed, we measured the distance from the short end to the crossing of the strands (*Figure 7—figure supplement 2c*), the size of the loops (*Figure 7—figure supplement 2d*), and the sum of these two measurements, referred to as the total length (*Figure 7—figure supplement 2e*). This analysis showed that the average distance from the short end of the molecule to the start of the loop was 245 ± 22 bp, and the average loop size was 409 ± 47 bp (*Figure 7—figure supplement 2f*). A representative image illustrating a loop is shown in *Figure 7c*. Interestingly, the position of the loop and its overall size are consistent with the entire transcription control region being located in the loop (*Figure 7c*). The sequence of the average loop is highlighted in yellow in *Figure 7d*. To determine more precisely the location of the DNA loops and to provide a control experiment that also tests the APTES liquid AFM imaging method (*Murugesapillai et al., 2017a*), we also developed a Ni$^{2+}$-mediated DNA-protein complex AFM imaging method. In this method, the DNA equilibrates in two dimensions (*Figure 7—figure supplement 3*), similar to the standard Mg$^{2+}$-mediated DNA imaging method. However, this method also allows imaging in liquid. Here we prepared a mtDNA or puc18 construct with a streptavidin label on one end, allowing us to identify the orientation of the DNA sequence. After again adding 10 nM TFAM, we obtained DNA-protein complex images, as shown in *Figure 7e* and *Figure 7—figure supplement 4*. The images allow us to clearly identify protein-mediated loops. The locations of each loop are shown in *Figure 7—figure supplement 5*. For mtDNA, we find 39% of the DNA loops begin and end in the IPR region, compared to only 18% for the same region on puc18. These results suggest a preference for looping near the IPR region for the mtDNA construct. Because loops also form at other locations, this preference is likely due to preferential, but not exclusive, binding near the IPR region. These results show that there is preferential looping near the IPR region and that a 2D DNA equilibration method also gives similar TFAM-mediated DNA looping results.

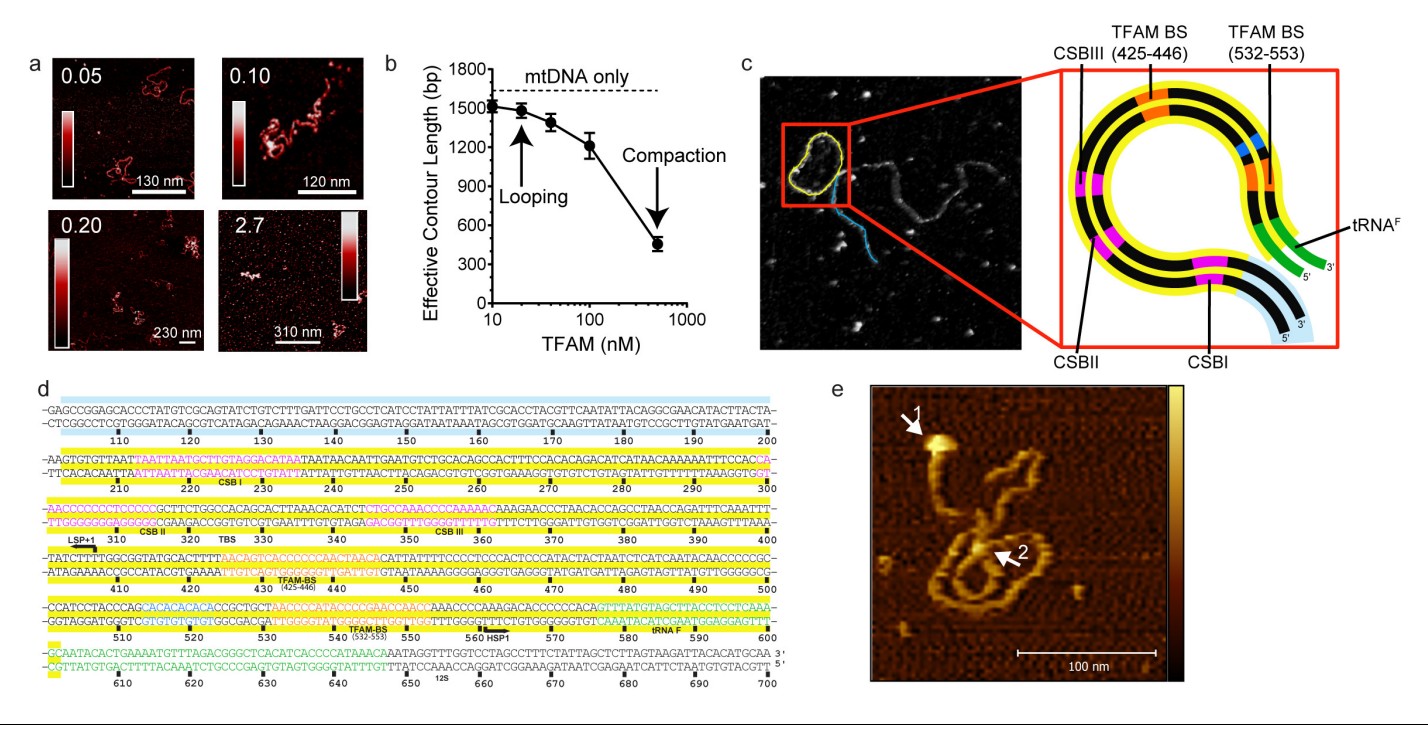

**Figure 7.** mtDNA looping correlates with activation and compaction with repression. (**a**) Representative AFM images of TFAM-mtDNA interaction as a function of TFAM concentration per bp, obtained in liquid. The ratio of TFAM/bp is indicated in the upper left corners of each AFM image. The color bar represents the sample height ranging from 0.0 to 2.0 nm. (**b**) Effective contour length of the IPR DNA as a function of TFAM concentration. The black dotted line represents the contour length of mtDNA in the absence of protein [5]. The filled circles (with connecting black line) represent the measured effective contour length of the mtDNA in the presence of TFAM. (**c**) The primary loop observed from AFM analysis of TFAM-mtDNA interaction is between the regions that encompass CSB1 to tRNA[F] and contains the three mitochondrial DNA promoters (LSP, HSP1 and HSP2) and the interpromoter region. In the greyscale image, the loop formed is illustrated in yellow and the region preceding the loop is illustrated in blue. (**d**) mtDNA sequence. Shown here is a small region of mtDNA sequence surrounding and including the IPR (#101–700; numbering according to the standard Cambridge mtDNA sequence). The yellow shading indicates the region of mtDNA sequence that gives rise to the primary loop observed using AFM. (**e**) The AFM image shows an individual mtDNA molecule with a streptavidin DNA-end label and a TFAM-wt mediated loop. The bound streptavidin is clearly visible at the biotin-tagged end of the mtDNA (white arrow no. 1). The height profile shows a height of 3.5–4 nm for the TFAM mediated loop at the mtDNA crossing (white arrow no. 2). The mtDNA itself shows a height of ~2 nm which is consistent with the height expected for low force AFM imaging of DNA in liquid.

The following figure supplements are available for figure 7:

**Figure supplement 1.** Imaging of mtDNA using AFM in liquid.

**Figure supplement 2.** TFAM-induced looping characterized by AFM in liquid.

**Figure supplement 3.** NiCl$_2$-mediated DNA-protein complex imaging method.

**Figure supplement 4.** NiCl$_2$ mediated DNA-protein complex deposition on Mica.

**Figure supplement 5.** NiCl$_2$ mediated DNA deposition on Mica.

**Figure supplement 6.** Distance to loop and distance to protein location.

**Figure supplement 7.** Percent of DNA molecules (either mtDNA or puc18) in each of four categories: Loop present but no detectable protein bound at the DNA crossover, loop present with protein bound at the crossover, no loop with protein bound somewhere on the molecule, and no loop and no detectable protein bound.

**Figure supplement 8.** Fraction of the DNA contour that is covered by protein as a function of protein concentration.

*Figure 7 continued on next page*

*Figure 7 continued*

**Figure supplement 9.** Characterization of the size of the protein complexes.

**Figure supplement 10.** Distribution of protein complex volume when bound to DNA.

To further link looping with transcriptional activation we performed quantitative analysis to demonstrate that sequence-specific binding by TFAM leads to protein-mediated looping. To do that, we used our AFM images to locate the position and size of each detectable protein cluster on the DNA, again using APTES-mediated AFM imaging (*Figure 7—figure supplement 6*). From this information, we quantitatively characterize the DNA and protein configurations for both mtDNA and the puc18 control construct. This experiment required interpretation of each observed crossover. Three types of crossovers can be identified: protein-bound DNA (purple arrow in *Figure 7—figure supplement 7b*); TFAM-mediated DNA bridges (blue arrow in *Figure 7—figure supplement 7b*); and naked DNA crossovers (green arrow in *Figure 7—figure supplement 7b*). Each of these conformations exhibits a diagnostic height in the AFM image (*Figure 7—figure supplement 7b*). We measured the height of the crossing of two DNA strands and we found the height to be $1.33 \pm 0.16$ nm (N = 12). For DNA only we found an average height of $1.10 \pm 0.15$ nm (N = 13). This illustrates that when DNA is imaged under these conditions, the crossing of two strands is not necessarily twice the height of DNA alone. However, we also calculated the height of proteins bound at DNA crossovers and we found a height of $1.95 \pm 0.12$ nm (N = 13). The errors are standard error of the mean. These data are consistent with loop formation resulting from sequence-specific recruitment of TFAM to mtDNA. *Figure 7—figure supplement 7c and d* show the percent of DNA molecules (either mtDNA or puc18) in each of four categories: Loop present but no protein, loop present with protein bound at the DNA crossover, no loop with protein bound, and no loop and no protein bound. For mtDNA, we see a strong peak in the category of loop formed with protein bound, while for puc18 these categories are all equally probable. This supports the preferential binding of TFAM to mtDNA and demonstrates how this preference leads to protein-mediated loops at the demonstrated concentration of 10 nM protein added. To fully quantify the binding affinity for these constructs, we also find the fractional binding of TFAM on each DNA lattice by determining the fraction of the DNA contour that is covered by protein as a function of protein concentration. *Figure 7—figure supplement 8* shows the measured fractional binding as a function of effective concentration of TFAM in solution (taking into account the reduction in solution concentration due to DNA binding). The results clearly demonstrate a factor of 3 increase in TFAM binding affinity for the mtDNA construct ($K_d = 59 \pm 2$ nM) relative to the puc18 construct ($K_d = 200 \pm 40$ nM) under our solution conditions. Because TFAM is a DNA bending protein, such an affinity increase due to the IPR sequences located in the loop could easily lead to nucleation of protein complexes at the loop crossing point. Thus, sequence-specific binding within the loop leads to protein binding at crossover points, even when these crossover points are not the preferred sequence.

Since AFM images allow us to visualize protein bound to DNA directly, we also characterized the size of the protein complexes. We first measured the volume distribution of individual proteins (*Figure 7—figure supplement 9a*), revealing an average volume of 61 nm$^3$ *Figure 7—figure supplement 9b*, similar to the calculated volume of 52 nm$^3$, and an average height of 0.98 nm (*Figure 7—figure supplement 9c*). We next measured the distribution of protein complex volume when bound to DNA. We find that the average volume of protein at the DNA crossing is $138 \pm 15$ nm$^3$, or 2 to 3 proteins. When the complexes are found outside the crossing point, the average volume is $103 \pm 9$ nm$^3$, or 1 to 2 proteins (*Figure 7—figure supplement 10*). These measurements support the hypothesis that crossing points nucleate TFAM cluster formation, consistent with the model presented above. The cluster sizes observed are consistent with previous measurements of the cooperativity of TFAM binding ($\omega = 70$, [*Farge et al., 2012*]). This further demonstrates that the protein clusters found at loop crossing points are on average dimers, providing quantitative data for multimerization of TFAM at 10 nM, a concentration shown to be biochemically relevant in transcription assays (*Figure 7—figure supplement 10*).

We were then in a position to determine the importance of the TFAM carboxy-terminal tail for DNA looping. Under conditions in which looped mtDNA molecules were prevalent in the presence of TFAM, looped mtDNA molecules were rare in the presence of TFAM-CTΔ26 (*Figure 8a*). In fact, only 24% (N = 29) of molecules showed any loop-like conformations, similar to mtDNA alone (*Figure 8b*). A trivial explanation for this observation is that TFAM-CTΔ26 does not bind to DNA in the range from 10 to 100 nM. To test this, we used AFM imaging in liquid to determine the extent of TFAM-CTΔ26 binding under the conditions of this study. Because the persistence length of mtDNA decreases upon TFAM binding, persistence length can be used to determine if and when TFAM-CTΔ26 is bound to mtDNA (*Murugesapillai et al., 2014*; *Rivetti et al., 1996*; *et al., 2009*). As expected, a value of 50 ± 2 nm was measured in the absence of TFAM (*Figure 8d*). The persistence length decreased to 30 ± 2 nm in the presence of 10 nM TFAM (*Figure 8d*), consistent with previous measurements (*Farge et al., 2012*). The persistence length decreased only to 46 ± 1 nm in the presence of 10 nM TFAM-CTΔ26, consistent with a somewhat lower binding affinity to mtDNA than TFAM (*Figure 8d*). In contrast, the persistence length of mtDNA decreased to 33 ± 1 nm in the presence of 100 nM TFAM-CTΔ26, similar to that observed for 10 nM TFAM (*Figure 8d*), suggesting that TFAM-CTΔ26 has approximately ten-fold weaker binding to mtDNA relative to TFAM. To determine if this decrease in binding affinity could be responsible for the observed inability of TFAM-CTΔ26 to induce mtDNA looping, we also measured the looping fraction at 100 nM TFAM-CTΔ26, and the looping probably was still similar to that observed for mtDNA only (*Figure 8c*). We therefore conclude that the lack of looping enhancement by TFAM-CTΔ26 is due to the properties of the carboxy-terminal tail and not due to reduced DNA binding affinity upon removal of the carboxy-terminal tail. The efficiency with which TFAM-CTΔ26 condenses mtDNA into nucleoids is reduced comparably with the reduced efficiency of DNA binding (*Figure 8—figure supplement 1c*). Interestingly, compaction occurs in the absence of loops for TFAM-CTΔ26 (*Figure 8—figure supplement 1c*), thus ruling out loops as obligatory intermediates in the mechanism of DNA compaction by TFAM.

To confirm that TFAM-induced DNA looping occurs in the absence of a surface, we performed optical-tweezers experiments (*Chaurasiya et al., 2010*; *Murugesapillai et al., 2014*). In such experiments, a 48.5 kbp DNA is tethered at each of its termini to beads; one bead is held in an optical trap while the other is held on a glass micropipette. At low force and extension [less than one pico-Newton (pN)], the DNA is free to sample multiple conformations. A force can be exerted to pull one bead away from the other and extend the tethered DNA. In the absence of TFAM, the force-extension curve was smooth (*Figure 9a*). However, in the presence of TFAM, jumps were present in the force-extension curve, consistent with the force breaking loops created by TFAM dimers (*Figure 9b*). Loops were readily detected when the DNA was extended at a rate of 970 nm/s but not when the DNA was extended at a rate of 100 nm/s. The DNA was relaxed and immediately extended, yielding a force-extension curve resembling naked DNA (*Figure 9c*). However, when the DNA was relaxed, held for several minutes, and then extended, a jump in the force-extension curve was apparent (*Figure 9d*). These data suggest that TFAM dimerization likely stabilizes spontaneously formed DNA loops. Finally, we determined the strength of the TFAM dimer by measuring the average force required to break a loop (*Figure 9e*). This value was 17 ± 2 pN (N = 15). This is similar to the force required to break loops formed by the yeast HMG-box protein, HMO1 (*Murugesapillai et al., 2014*).

## Discussion

The pioneering work of the Attardi and Clayton laboratories identified three transcription start sites on mitochondrial DNA, one on the light strand and two on the heavy strand (*Chang and Clayton, 1984*; *Montoya et al., 1982*, *1983*). These studies also revealed that transcription from the first heavy-strand promoter (HSP1) produced far more RNA than transcription from the second heavy-strand promoter (HSP2) (*Montoya et al., 1983*). Collectively, this early work, much of which was performed using cell-based and/or cell-free experiments, made a good case for differential regulation of mitochondrial transcription. Fast forward 30 years to the current era of studying mitochondrial transcription using purified components and a case has now been made for only two promoters: LSP and HSP1, and the absence of differential regulation (*Litonin et al., 2010*; *Morozov and Temiakov, 2016*; *Shi et al., 2012*). The goal of this study was to resolve this paradox.

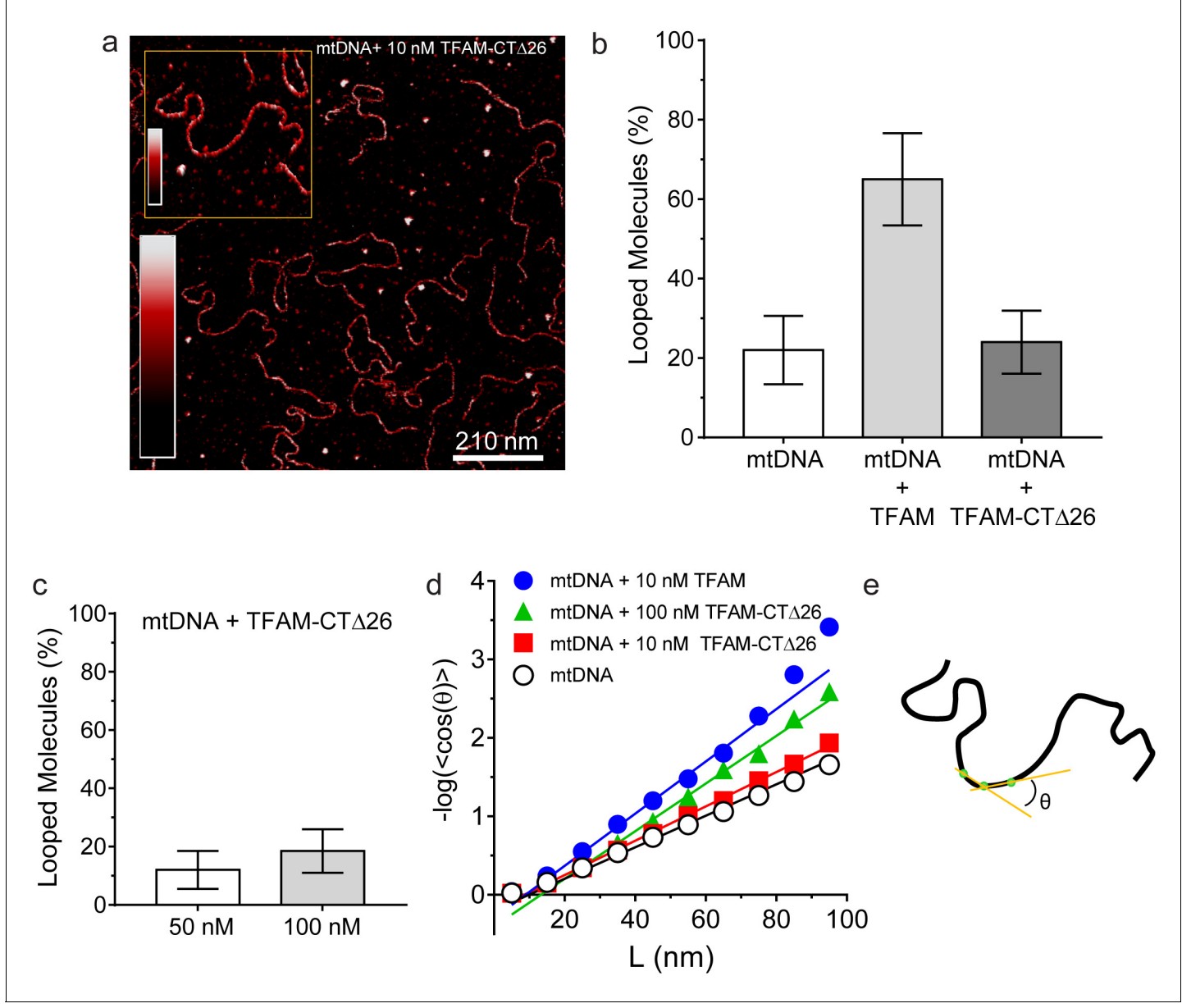

**Figure 8.** DNA looping is enhanced in the presence of TFAM, but not in the presence of TFAM-CTΔ26, even when strongly bound. (a) Two-dimensional representation of mtDNA bound to TFAM-CTΔ26. The inset shows a three-dimensional representation of a selected region from the same image. The color bars represents the sample height ranging from 0.0 to 2.0 nm. (b) The bar graph shows the percentage of looped molecules that were observed for mtDNA constructs in the absence and presence of 10 nM TFAM or TFAM-CTΔ26. (c) Percentage of looped mtDNA molecules in the presence of 50 nM and 100 nM TFAM-CTΔ26, showing that even when strongly bound, TFAM-CTΔ26 does not increase DNA looping. (d) Fits to the 3D wormlike chain model for mtDNA construct in the absence and presence of TFAM or TFAM-CTΔ26. The results from the fits give persistence lengths of 50 ± 2 nm for mtDNA alone, 30 ± 2 nm for mtDNA with 10 nM TFAM, 46 ± 2 nm for mtDNA with 10 nM TFAM CTΔ26, and 33 ± 1 nm for mtDNA with 100 nM TFAM CTΔ26. These results show that 100 nM TFAM CTΔ26 is equivalent in DNA binding to 10 nM TFAM. (e) Diagram of a single DNA molecule. The distance between two consecutive green dots is L. The angle formed between these adjacent segments is θ.

The following figure supplement is available for figure 8:

**Figure supplement 1.** Both TFAM and TFAM-CTΔ26 compact DNA at high concentrations.

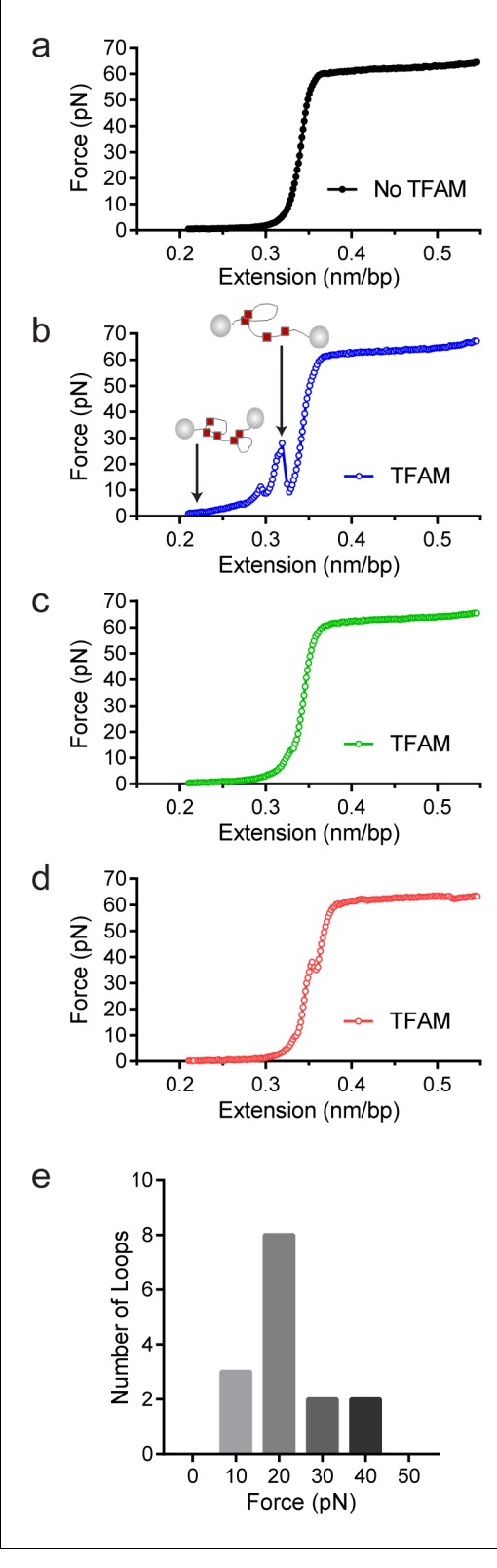

**Figure 9.** Optical tweezers data confirm DNA looping by TFAM. (**a**) Force-extension curve of bacteriophage-λ molecule of 48,500 base pairs (bp) in the absence of TFAM. (**b**) Initial force-extension curve of
*Figure 9 continued on next page*

Studies that were unable to observe differential regulation of LSP and HSP1 in reconstituted systems shared several features of concern (*Litonin et al., 2010*; *Shi et al., 2012*). First, 'minimal' promoters were used. Second, reaction products were labeled with a radioactive nucleotide, a condition requiring distorted nucleotide concentrations and associated caveats. Third, absolute transcript quantification was not performed, so it was unclear how many initiation events were being monitored or if all templates were (re)used. In this study, we address all of these concerns. We show that transcription from HSP1 can occur in the absence of TFAM (*Figure 1*). Quantitative analysis shows that in the absence of TFAM only one transcript is produced from HSP1 per template (*Figure 1* and *Figure 1—figure supplement 3*). Thus, the inability of others to detect a single round of transcription would explain their inability to observe the TFAM-independent transcription from HSP1.

TFAM functions differently at LSP and HSP1 (*Figure 10*). TFAM is essential for transcription initiation at LSP (*Figure 1*). Here TFAM functions stoichiometrically with the template and supports multiple rounds of reinitiation from LSP as 5–10 LSP transcripts are produced from each template (*Figures 1* and *2*). In contrast, TFAM appears to be essential only for reinitiation from HSP1 (*Figure 1*). We suggest that the nascent transcript may not be efficiently displaced in the absence of TFAM. In the absence of TFAM, regardless of the amount of DNA upstream or downstream of the HSP1 transcription start site and/or the duration of the incubation, transcript yield was always stoichiometric with template (*Figures 1*, *2* and *6*). The first 20 nt of the HSP1 transcript is GC rich (see 561–581 in *Figure 7d*), in contrast to the AT-rich LSP transcript. Perhaps the failure of nascent RNA to be displaced leads to formation of an R-loop (*Wanrooij et al., 2012*, *2010*). It is well known that POLRMT elongation can be impeded by stretches of guanylate residues, for example conserved sequence box II (CSB II, 299–315 in *Figure 7d*), perhaps because of their propensity to form quadruplexes (*Wanrooij et al., 2012*, *2010*). Arrest at CSB II can be prevented by the mitochondrial transcription elongation factor, TEFM (*Agaronyan et al., 2015*). TFAM may prevent arrest at guanylate stretches during initiation. Deep-sequencing approaches that permit identification of sites of transcription pausing and/or arrest identified the region be positions 600 and 700 of the human mtDNA as sites of POLRMT pausing or arrest (*Blumberg et al., 2017*).

*Figure 9 continued*

bacteriophage-λ molecule in the presence of 50 nM TFAM (blue open circles). When held at low extension loops are mediated in the presence of TFAM and as the molecule of DNA is extended we observe jumps revealing the breaking of a loop previously formed. The cartoon inset illustrates the formation of loops mediated by TFAM and the breaking of loops as the DNA is extended. (**c,d**) Consecutive force-extension curves in the presence of TFAM are shown in green (panel c, extended immediately after the initial extension shown in panel b) and red (panel d, extended after waiting 7 min). (**e**) The histogram illustrates the forces involved in breaking the loops mediated by TFAM. The most probable loop breaking force is 20 pN.

The carboxyl terminus of TFAM is essential for its transcription-activation function at HSP1 but not at LSP (*Figure 5*). We and others have previously made similar observations (*Lodeiro et al., 2012*; *Ngo et al., 2014*); however, a debate remains (*Morozov and Temia-kov, 2016*). We now show that production of more than one transcript from HSP1 is impossible when the TFAM is truncated by removal of its carboxy-terminal 26 amino acid residues, though there is no impact on the first round of transcription from HSP1 (*Figure 5*). In contrast, transcription from LSP is only modestly reduced (*Figure 5*). Transcription repression activity of TFAM is unaltered by loss of the carboxyl terminus (*Figure 5*). The carboxy-terminal tail has been shown to contribute to intermolecular interactions that could give rise to DNA looping (*Figure 5c*) (*Ngo et al., 2011*; *Rubio-Cosials et al., 2011*). DNA looping is a well-established paradigm for transcription activation in prokaryotes (*Cournac and Plumbridge, 2013*). Looping could contribute to displacement of nascent RNA by inducing sufficient strain in the template to prevent stable hybridization of the transcript, thus facilitating multiple rounds of initiation at HSP1. If the role of looping is to increase yield from HSP1, then post-translational modifications in the carboxy-terminal tail could regulate loop formation and therefore transcriptional output from HSP1. While numerous post-translational modifications of TFAM have been observed (*Grimsrud et al., 2012*), none of these map to the carboxy-terminal tail. It should be noted that most of the proteomic studies have been performed using rapidly-dividing cancer cell lines; mitochondrial transcription and biogenesis are quite active under these conditions.

TFAM-mediated looping of pUC19 plasmid DNA has been observed using atomic force microscopy (AFM) imaging in air (*Kaufman et al., 2007*), although this result was later suggested to be due to random crossovers rather than TFAM-mediated looping (*Farge et al., 2012*). Here we show using both AFM imaging in liquid and optical tweezers that TFAM actively mediates DNA looping, and our AFM studies show that it creates loops on mtDNA sequences as well (*Figure 7c*). The average size of the loops formed on mtDNA sequences is larger than observed on plasmid DNA (*Figure 7—figure supplement 2*), and more loops are found per molecule with mtDNA sequences than with plasmid DNA (*Figure 7—figure supplement 7*). TFAM-mediated loops appear to be localized to the control region, and most loops include HVR3 (*Figure 7—figure supplement 2*). Looping is strictly dependent on the carboxy-terminal tail of TFAM (*Figure 8*). Interestingly, the inability to form loops has little impact on the ability of TFAM to compact DNA (*Figure 8—figure supplement 1*), suggesting that loops are not obligatory intermediates on the path for compaction as suggested previously (*Kaufman et al., 2007*). We are intrigued by the possibility that TFAM-mediated looping represents another layer of transcriptional regulation (*Figure 10*). TFAM binding induces a U-turn into the DNA at LSP (*Ngo et al., 2011*; *Rubio-Cosials et al., 2011*). At HSP, in addition to a U-turn, looping is required.

A particularly unexpected outcome of this study is the finding that sequences as far as 114 bp upstream (*Figure 2*) and 95 bp downstream (*Figure 6*) of the HSP1 transcription start site contribute to the transcriptional output of HSP1. Collectively, our data suggest that TFAM binds within HVR3 and the tRNA$^F$ gene to create loops that are used to facilitate transcription. TFAM protection of HVR3 is readily observed by DNAse I footprinting (*Figure 4*, *Figure 4—figure supplement 1*); however, the footprint is lost completely when the TFAM carboxy-terminal tail is deleted (*Figure 5—figure supplement 1* and *Figure 5—figure supplement 2*).

Dogma in the field has been that transcription initiation from LSP and HSP1 is regulated by similar or identical mechanisms (*Litonin et al., 2010*). Because of this perspective, there has been an effort to force HSP1 to fit the paradigm established for LSP. One example is the widely-held belief that the TFAM-binding site of HSP1 (referred to here as region five and shown in red at 532–553 in

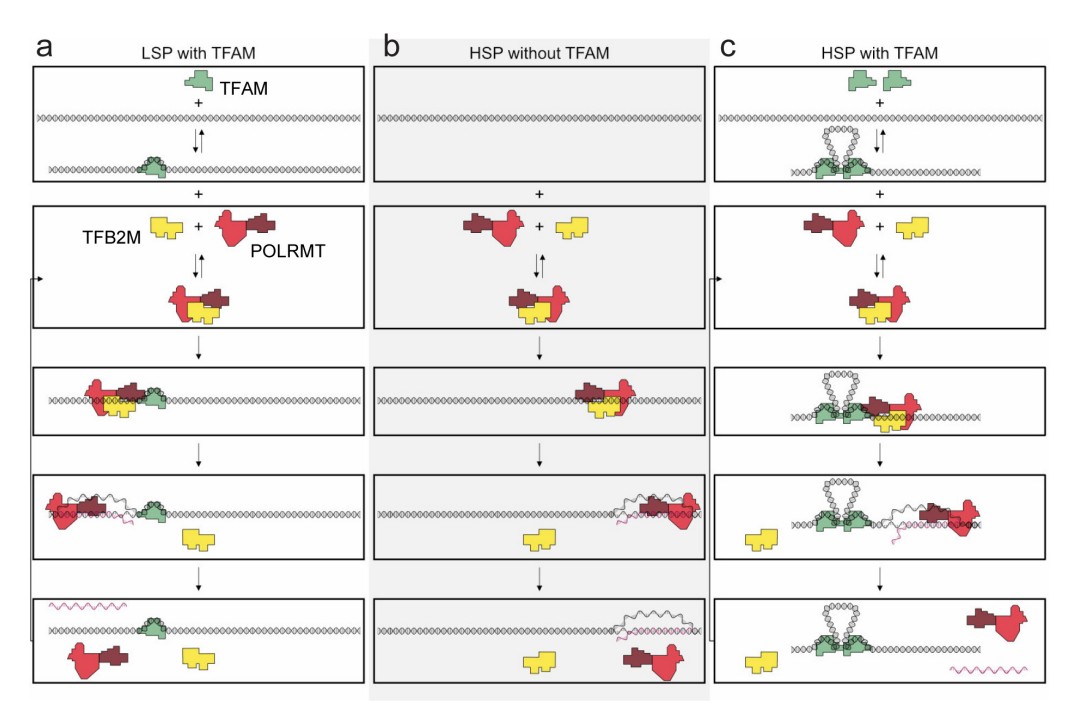

**Figure 10.** Regulation of mitochondrial transcription by TFAM. (a) Initiation of transcription at light-strand promoter (LSP) requires TFAM. A TFAM monomer binds to a specific site upstream of the LSP transcription start site (TSS) and bends the DNA. POLRMT and TFB2M associate and add to LSP, perhaps directed by TFAM. Initiation requires all three components at this promoter; elongation only requires POLRMT. Elongation to the end of template leads to dissociation of POLRMT and RNA product from template, thus enabling another round of transcription. (b) Initiation of transcription at heavy-strand promoter 1 (HSP) does not require TFAM. POLRMT and TFB2M associate and are sufficient to recognize and bind to HSP. Initiation requires only these two components, and elongation requires only POLRMT. Elongation to the end of template leads to dissociation of POLRMT, but additional rounds of transcription are not supported, perhaps because RNA product remains hybridized to template. (c) Reutilization of HSP requires TFAM. Binding of a TFAM dimer to the inter-promoter region creates loops of the DNA. Formation of the loops requires the carboxy-terminal tail of TFAM. POLRMT and TFB2M associate and add to TFAM-bound HSP. Initiation and elongation occur as described above; except TFAM facilitates multiple turnovers from HSP. Looping is required for this function as deletion of the carboxy-terminal tail precludes TFAM activation at HSP. FIGURE SUPPPLEMENTS.

The following figure supplement is available for figure 10:

**Figure supplement 1.** SNPs and somatic mutations associated with disease in LSP, HSP1 and HVR3.

*Figure 7d*) resides approximately 50 bp upstream of the start site, as does the TFAM-binding site of LSP (referred to here as region one and shown in red at 425–446 in *Figure 7d*). Additionally, these two regions (regions 1 and 5) are similar in sequence but are in reverse orientations relative to their transcription start sites (*Fisher et al., 1987*). This has been suggested to position the carboxy-terminal tail in opposite orientations relative to the transcriptional machinery at the transcription start sites (*Ngo et al., 2014*). Here we show that region five functions differently than region 1. Binding of region five by TFAM occurs at concentrations of TFAM in which transcription from HSP1 is inhibited (*Figure 4* and *Figure 4—figure supplement 1*). Alterations of region five converts TFAM into a potent repressor of transcription from HSP1 (*Figure 3*). Repression by TFAM requires its carboxy-terminal tail (*Figure 5*). It is possible that region five has evolved to prevent TFAM binding, bending and looping from occluding the site used by POLRMT and TFB2M to bind to the promoter. If this is the case, then it might also be possible to use DNA modifications, for example cytosine methylation or even guanine oxidation, of region five to control transcription from HSP1.

Our studies provide the first suggestion for the role of HVR3, the LSP-HSP1 inter-promoter region (IPR), in mitochondrial transcription. We suggest that HVR3 sequences from 450 to 560 are an integral component of HSP1. The transcriptional function of HVR3 is mediated by TFAM binding.

Footprinting demonstrates that binding of TFAM to HVR3 is not random (*Figure 4*). After binding, TFAM-TFAM interactions lead to the formation of loops that permit multiple rounds of transcription to occur from HSP1. Somatic mutations in HVR3 have been linked to diseases, such as cancer (*Figure 10—figure supplement 1*). The mechanism is completely unknown. Our studies suggest that these mutations could possibly alter mitochondrial gene expression. Also worth noting is the fact that many of the somatic mutations linked to disease actually are present as polymorphisms in 'normal' individuals (*Figure 10—figure supplement 1*). It is possible that disease-associated polymorphisms predispose to disease. We conclude that natural variation in HVR3 may lead to between-individual differences in mitochondrial transcription or, in the worst cases, misregulated mitochondrial transcription that contributes to disease.

In conclusion, the experiments described here demonstrate differential regulation of transcription (re)initiation from LSP and HSP1 mediated by TFAM (*Figure 10*). Initiation at LSP likely begins with TFAM binding and bending promoter DNA followed by recruitment of POLRMT and TFB2M, likely already in complex. In the presence of nucleoside triphosphates, di- and/or tri-nucleotide products are produced and further extended into full-length RNA. LSP templates are efficiently reutilized and factors recycled. Initiation at HSP can occur with POLRMT and TFB2M only. However, after production of transcript, the template fails to be reutilized. As stated above, this could be due to formation of R-loops, triplexes or some other heteroduplex. Additional studies will be required to understand this phenomenon. In the presence of TFAM, HSP1 template reutilization becomes feasible. We attribute template reutilization to the ability of TFAM to form loops in a manner dependent on its carboxy-terminal tail. The molecular basis for this property of TFAM-mediated loops remains unclear. Collectively, these studies reveal clear differences in the cis-acting elements and trans-acting factors required for transcription initiation from LSP and HSP1. These differences are presumably exploited for differential promoter regulation, and may play roles in disease.

## Materials and methods

### Materials

The proteins used to perform this study can be obtained from Indigo Biosciences (State College, PA). DNA oligonucleotides were from Integrated DNA Technologies. Oligonucleotide primers used in this study are listed in *Table 1*. RNA oligonucleotides were obtained from Dharmacon. Restriction endonucleases were from New England Biolabs. T4 polynucleotide kinase was from USB. RQ1 DNase was from Promega. Ultrapure NTP solutions were from GE Healthcare. $[\gamma\text{-}^{32}P]ATP$ (7000 Ci/mmol) was from Perkin Elmer. HPLC purified $[\gamma\text{-}^{32}P]$ ATP (3000 Ci/mmol) was from MP Biomedical. 10 bp DNA ladder was from Invitrogen. All other reagents were of the highest grade available from Sigma, Fisher or VWR.

### Preparation of dual promoter template

Dual promoter DNA templates were prepared by PCR. The D-loop region of mtDNA (CRS, NC_012920) was cloned into pUC18 and used as template for PCR. Seven 100 μL reactions contained final concentration of 0.5 ng/μL plasmid template, 1 μM LSP +35 For primer, 1 μM HSP1 +45 Rev primer, 0.3 mM dNTP, 1 × Thermo Pol Buffer (NEB), and DeepVent DNA polymerase (NEB). Dual promoter deletion, inverted, randomized and AFM constructs were also prepared by PCR but using the appropriate forward and reverse primers listed in *Table 1*. PCR cycling conditions are as follows: 1 cycle at 95°C for 4 min; 40 cycles of denaturing at 95°C for 30 s, annealing at 57°C for 30 s, and extension at 72°C for 20 s; 1 cycle at 72°C for 10 min. PCR products were purified with Wizard SV gel and PCR Clean-up System (Promega), DNA was eluted in 80 μL TE (10 mM Tris-HCl, pH 8.0 buffer and 0.1 mM EDTA) and finally diluted to 1 μM in TE buffer. Extinction coefficients for each DNA construct were calculated with IDT DNA technologies tool (http://biophysics.idtdna.com/UVSpectrum.html).

### In vitro transcription assays

Reactions were performed in 1 × reaction buffer (10 mM HEPES pH 7.5, 100 mM NaCl, 10 mM MgCl₂, 1 mM TCEP and 0.1 μg/μL BSA) with 10 μM $^{32}P$-end-labeled RNA primer (pAAA), 500 μM NTP and 100 nM DNA template. Reactions were performed by incubating template DNA in reaction

**Table 1.** DNA Oligonucleotides used in this study.

| DNA oligo | Sequence |
| --- | --- |
| LSP + 35-For | 5'AACACCAGCCTAACCAGATTTC3' |
| HSP1 + 45-Rev | 5'ATTGCTTTGAGGAGGTAAGC3' |
| HSP1-138-For | 5'TTAACAGTCACCCCCCAACTAAC3' |
| HSP1-114-For | 5'CATTATTTTCCCCTCCCACTC3' |
| HSP1-60-For | 5'CCATCCTACCCAGCACACAC3' |
| HSP1-50-For | 5'CAGCACACACACACCGCTGC3' |
| LSP-146-Rev | 5'GGTTGGTTCGGGGTATGGG3' |
| LSP-115-Rev | 5'GTGTGTGTGCTGGGTAGGATG3' |
| LSP-86-Rev | 5'TTGTATTGATGAGATTAGTAG3' |
| LSP-69-Rev | 5'GTAGTATGGGAGTGGGAGG3' |
| LSP-40-Rev | 5'GTGTTAGTTGGGGGGTGAC3' |
| IPR-Reg2-For | 5'ATTATTTTCCCCTCCCACTCCC3' |
| HSP1 + 70-Rev | 5'GAGCCCGTCTAAACATTTTC3' |
| HSP1 + 95-Rev | 5'AACCTATTTGTTTATGGGGTG3' |
| HSP1 + 120-Rev | 5'AGAGCTAATAGAAAGGCT3' |
| HSP1 + 145-Rev | 5'GATGCTTGCATGTGTAATCTTACTAAGAGCTAAT3' |
| °-Reg1-For | 5'GCACTTTTATTATTTTCCCCTCCCAC3' |
| °-Reg1-Rev | 5'GGAAAATAATAAAAGTGCATACCGCC3' |
| °-Reg5-For | 5'CAGCACACACACACCAAACCCCAAAGACAC3' |
| °-Reg5-Rev | 5'GGTTTGGTGTGTGTGTGCTGGGTAG3' |
| Inv-Reg5-For | 5'TGGTTCGGGGTATGGGGTTAGCAGCGGTACCAAACCCCAAAGACACCC3' |
| Inv-Reg5-Rev | 5'ACCGCTGCTAACCCCATACCCCGAACCAGTGTGTGTGCTGGGTAGGATGG3' |
| Rdm-Reg5-For | 5'GACCCGACCTCCCTACCACCACTCCGAACGACCACCCACCAAACCCCAAAGACACCCCC3' |
| Rdm-Reg5-Rev | 5'TGGTGGTAGGGAGGTCGGGTCGGGGTGTATGAGGTGGGTCTTTGTATTGATGAGATTAGTAGTATGG3' |
| LSP + 510-For | 5'CTTCCGGCTCGTATGTTGTGTGGAATTG3' |
| HSP1 + 1000-Rev | 5'ACTTTAAAAGTGCTCATCATTGG3' |

buffer at 32°C for 5 min and then adding in the following order: TFAM (varying concentrations), TFB2M and POLRMT (1 μM each). Between each addition of protein to the reaction there was an incubation time of 1 min. After addition of POLRMT, the reaction was allowed to incubate at 32°C for 5 to 60 min. At each time point 4 μL of the reaction mix were quenched into 8 μL of stop buffer (79.2% formamide, 0.025% bromophenol blue, 0.025% xylene cyanol and 50 mM EDTA final). Products were resolved by denaturing 20% (37:3, acrylamide:bis-acrylamide ratio) PAGE. Proteins were diluted immediately prior to use in 10 mM HEPES, pH 7.5, 1 mM TCEP, and 20% glycerol. The volume of protein added to any reaction was always less than or equal to one-tenth the total volume. Any deviations from the above are indicated in the appropriate figure legend. Gels were visualized by using a PhosphorImager (GE) and quantified by using ImageQuant TL software (GE) (RRID:SCR_014246).

## DNAse I footprinting

Footprinting reaction probes of dual promoter template were prepared as described above except that the PCR was performed with either LSP +35 For or HSP1 +45 Rev primer which were [32]P-labeled prior to PCR. DNAse I footprinting was performed in the presence of TFAM alone or in the presence of TFAM, TFB2M and POLRMT. Reaction buffer consisting of 10 mM HEPES, pH 7.5, 10 mM $MgCl_2$, 10 mM $CaCl_2$, 100 mM NaCl, 1 mM TCEP, and 0.1 mg/mL BSA at final concentration was prepared and incubated for 1 min at 32°C prior to addition of proteins in the following order:

TFAM (varying concentrations), TFB2M (1 µM) and POLRMT (1 µM). Between each addition of protein to the reaction there was an incubation time of 1 min. Upon addition of all proteins, the reaction was incubated for 5 min followed by adding mixture of RQ1 DNase (0.002 units/µL, final) and $CaCl_2$ (1 mM, final). The DNA digestion for 2 min at 32°C was quenched by addition of EDTA and urea at final 40 mM and 840 mM, respectively, in 5 µL. Proteinase K was mixed with each reaction at final 0.04 µg/µL, and incubated at 50°C for 15 min to digest TFAM and other proteins. Finally, 20 µL of loading buffer (79.2% formamide, 50 mM EDTA, 0.025% xylene cyanol and 0.025% bromophenol blue, and 5 µM trap oligonucleotide DNA, a 90 bp oligo of either the LSP or HSP strand, whichever was [32]P-labeled) were added to each reaction. Samples were heated at 95°C for 5 min prior to loading PAGE. Products were resolved by denaturing PAGE on 7% gels (37:3, acrylamide:bis-acrylamide ratio). Gels were visualized by using a PhosphorImager (GE) and quantified by using ImageQuant TL software (GE) (RRID:SCR_014246).

## Characterization of TFAM-DNA interactions using atomic force microscopy in liquid

### APTES-mediated deposition on mica

The sample was prepared using a freshly cleaved mica surface coated with 3-aminopropyl-trietoxy silatrane (APTES) (*Shlyakhtenko et al., 2013*). The mtDNA construct used for AFM was prepared by PCR as described above but with extension times of 2 min and was a fragment of mtDNA that began 510 bp downstream of LSP and 1000 bp downstream of HSP1. The concentration of mtDNA and pUC18 (pUC) used for the experiment was 0.11 nM. The concentration of protein ranged from 10 nM to 500 nM, corresponding to a ratio of protein to DNA varying from 0.05 to 2.7 [TFAM]/[bp]. The experiment was carried out using 25 mM NaCl, 25 mM Hepes at pH 7.5. We used a Bruker Nanoscope V MultiMode eight with PeakForce Tapping mode atomic force microscope (AFM) to image in liquid TFAM proteins bound to DNA, as previously described (*Murugesapillai et al., 2017a*). We use peak force as an imaging signal, in which force information is collected at every pixel of the image. To image in a liquid environment, a silicon cantilever was used (resonance frequency = 70 kHz, spring constant = 0.4 N/m and tip radius = 2 nm). Image processing was done using Nanoscope Analysis software. The scan range was varied from 1 µm X 1 µm to 2 µm X 2 µm at 512 × 512 pixels and at 1024 × 1024 pixels, respectively. The AFM images were quantified using NCTracer (RRID:SCR_000116), a software program developed by the Neurogeometry Lab at Northeastern University (*Chothani et al., 2011*; *Gala et al., 2014*). To determine the persistence length *p* the orientation differences θ along the DNA as a function of contour length L were fit to the 3D wormlike chain (WLC) model

$$\langle cos(\theta) \rangle = e^{-L/p} \tag{1}$$

For protein volume calculations (*Ratcliff and Erie, 2001*), the protein-bound DNA threshold was set at 1.5 nm and the volume was calculated using Gwyddion (http://gwyddion.net/).

### NiCl$_2$ mediated deposition on mica

The freshly cleaved mica surface was preloaded with 50 mM $NiCl_2$ buffer and washed with MilliQ-water after 10 min and the mica was dried gently under a stream of nitrogen (*Alonso-Sarduy et al., 2013*; *Piétrement et al., 2003*). The biotin-tagged mtDNA or pUC18 constructs (via biotinylated primers) were end-labeled by adding streptavidin (Sigma-Aldrich, St. Louis, MO) to the DNA solution, otherwise DNA constructs and TFAM were prepared as for APTES deposition as described above. Subsequently, TFAM was added to the solution and after 10 min incubation time the solution was mixed with $NiCl_2$ deposition buffer (25 mM NaCl, 25 mM Hepes, 1 mM $NiCl_2$ at pH 7.5) and loaded immediately onto the mica surface (*Piétrement et al., 2003*). After 10 min deposition time the surface was washed with buffer and imaged in buffer at room temperature (25 mM NaCl, 25 mM Hepes, 1 mM $NiCl_2$ at pH 7.5) using a Bruker Dimension Icon (Bruker Nano Inc., MO) with PeakForce Tapping mode. Bruker PEAKFORCE-HIRS-F-B AFM probes (f = 100 kHz, k = 0.12 N/m, Tip radius ~1 nm) were used to enable high resolution imaging of the DNA-protein complexes. The scan range was varied from 1 × 1 µm to 2 × 2 µm at 512 × 512 pixels. DNA segments were traced and analyzed with FiberApp software (*Usov and Mezzenga, 2015*). Similar to the 3D WLC model for the APTES

method, the 2D WLC model was used for the NiCl$_2$ deposition method for the persistence length calculation by fitting the orientation differences θ along the DNA as a function of contour length L

$$\langle cos(\theta) \rangle = e^{-L/2p} \tag{2}$$

## Characterization of TFAM-DNA interactions using optical tweezers

Optical tweezers were used to further investigate TFAM-DNA interactions. In this experiment, two high power laser beams are focused into a small spot of approximately 1 μm size, which acts as a trap for high refractive index beads compared to the surrounding medium. A single phage-λ DNA molecule (48,500 base pairs) labeled on the termini of its opposite strands with biotin can be attached by its termini to streptavidin-coated polystyrene beads (Bangs Labs). This allows TFAM-DNA interactions to be studied under tension (*McCauley et al., 2013*; *Murugesapillai et al., 2014*). The experiments were carried out in a buffer containing 10 mM Hepes, pH 7.5, and 100 mM Na$^+$. The forces are measured in picoNewtons and extension in nm/bp, the total DNA extension divided by the total number of DNA base pairs. To oberve and study loop formation in the presence of TFAM, DNA was brought to a very low extension at a force less than one picoNewton (*Murugesapillai et al., 2014*).

## Acknowledgements

We appreciate the thoughtful, constructive comments on various drafts of the manuscript by our colleagues. We thank Roel Fleuren for producing *Figure 10*. This study was funded, in part, by the endowment from the Eberly Family Chair in Biochemistry and Molecular Biology to CEC. LJM is supported by a grant from NIH, GM075965. MW and his laboratory are supported by a grant from NSF, MCB-1243883.

## Additional information

### Funding

| Funder | Grant reference number | Author |
| --- | --- | --- |
| National Institute of General Medical Sciences | GM075965 | James Maher |
| National Science Foundation | MCB-1243883 | Mark C Williams |
| Eberly Family Chair Endowment | | Craig E Cameron |

The funders had no role in study design, data collection and interpretation, or the decision to submit the work for publication.

### Author contributions

AU, DM, MFL, JJA, Conceptualization, Data curation, Formal analysis, Writing—original draft, Writing—review and editing; MK, Conceptualization, Data curation, Formal analysis, Methodology, Writing—original draft, Writing—review and editing; YW, SP, Data curation, Formal analysis, Writing—original draft, Writing—review and editing; GVO, Formal analysis; LJM, Conceptualization, Formal analysis, Funding acquisition, Writing—original draft, Writing—review and editing; MCW, CEC, Conceptualization, Formal analysis, Funding acquisition, Writing—original draft, Project administration, Writing—review and editing

### Author ORCIDs

Divakaran Murugesapillai, http://orcid.org/0000-0002-9445-0330
Markus Kastner, http://orcid.org/0000-0003-2662-9344
Guinevere V Oliver, http://orcid.org/0000-0003-1679-360X
Jamie J Arnold, http://orcid.org/0000-0002-2345-9776
Mark C Williams, http://orcid.org/0000-0003-3219-376X
Craig E Cameron, http://orcid.org/0000-0002-7564-5642

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
