## [Decision Letter]

Thank you for submitting your article "Unexpected sequences and structures of mtDNA required for efficient transcription from the first heavy-strand promoter" for consideration by *eLife*. Your article has been favorably evaluated by Kevin Struhl (Senior Editor) and three reviewers, one of whom is a member of our Board of Reviewing Editors. The following individual involved in review of your submission has agreed to reveal her identity: Claire Wyman (Reviewer #2).

The reviewers have discussed the reviews with one another and the Reviewing Editor has drafted this decision to help you prepare a revised submission. The reviewers were excited about the identification of new promoter elements in the human mtDNA, a finding that advances our understanding of mitochondrial transcription. However, they raised serious concerns about the quality and interpretation of the biophysical data.

Summary:

This study investigates if and how the human mitochondrial promoters are differentially regulated. The authors build on their previous work using a dual-promoter template that suggested variable regulation of transcription from LSP and HSP1. Not only do they recapitulate this finding here, they also show that there are novel, previously unappreciated, sequence elements that regulate expression from the promoters – HSP1 in particular. The fact that the hypervariable region 1 is important for transcription regulation has several disease-related implications. Additionally, they show that changing the sequence of the promoter region alters TFAM binding, which correlates to changes in transcription.

The observation that the C-terminal tail of TFAM is essential for transcription from HSP1 prompted the authors to use atomic force microscopy and optical trapping to study the relationship between DNA looping – another process known to be critically dependent on the C-terminal tail – and transcription and TFAM binding. They find that this looping is important for reusing the HSP1 site, a finding that has not been observed previously for mitochondria. Overall, this study answers important questions in the field and provides novel insight into the regulation of mitochondrial transcription.

Essential revisions:

1) The key concern raised by the reviewers is that the looping may be caused by nonspecific TFAM binding to DNA. The authors should provide quantitative data on TFAM DNA binding, such as binding affinities for specific sites of DNA versus non-specific DNA. The biophysical analysis presented shows more loops if mitochondrial DNA is used. There is no evidence, however, for loops specifically related to protein bound to the DNA sites between the promoters that were described to be important for transcription from the dual-promoter template. This DNA sequence is within the loop, an arrangement hard to link mechanistically to transcriptional activity measured. The study does not provide a clear link between the looping and the transcriptional activation. The lack of mechanistic connection between these two data sets, one testing specific protein DNA interactions in transcriptional activity and the other reporting on interactions that may be non-specific in nature is a weak point of the current work and its presentation.

2) Related to the point above: AFM images can visualise proteins bound to DNA directly, a strength of this type of imaging. Most images shown in this work do not indicate the presence of proteins except in rather large complexes, much larger than expected of a TFAM protein. The authors should present quantitative data on TFAM-DNA binding, information that should be easily extracted from the images:% DNA bound by protein, position of proteins along the contour of DNA, size of DNA-bound proteins.

3) The optical tweezers experiment is not well explained in the Materials and methods or Results sections. The concentration of TFAM is not given. Λ DNA is used, so isn't all binding non-specific? It is assumed that rupture events represent protein-protein dimer breakage whereas any number of other possibilities exist (e.g. protein-DNA). Statements about protein disassociation in time are presented without any (quantitative) analysis or reference to known properties. The data as presented do not add usefully to this story – the authors should carefully revise.

4) TFAM is stated to be a dimer based only on a weak interface observed in a crystal. The importance of the C-terminus is clear but the relevance of dimerization is not supported. The authors should provide or cite quantitative date for the multimerization of TFAM in biochemically relevant conditions.

5) The observation that subtle changes in TFAM concentration have significant effects on the looping/compaction of DNA and the differential transcription from the mitochondrial promoters is not new. Surprisingly, the authors make no mention of the 2014 Cell Reports paper by Farge and co-workers that provide insight that is highly relevant to this work.

---

## [Author Response]

*Essential revisions:*

*1) The key concern raised by the reviewers is that the looping may be caused by nonspecific TFAM binding to DNA. The authors should provide quantitative data on TFAM DNA binding, such as binding affinities for specific sites of DNA versus non-specific DNA. The biophysical analysis presented shows more loops if mitochondrial DNA is used. There is no evidence, however, for loops specifically related to protein bound to the DNA sites between the promoters that were described to be important for transcription from the dual-promoter template. This DNA sequence is within the loop, an arrangement hard to link mechanistically to transcriptional activity measured. The study does not provide a clear link between the looping and the transcriptional activation. The lack of mechanistic connection between these two data sets, one testing specific protein DNA interactions in transcriptional activity and the other reporting on interactions that may be non-specific in nature is a weak point of the current work and its presentation.*

We agree with the reviewers that the protein-DNA interactions studied using AFM must be sequence-specific to be relevant to transcriptional activity measured, and that we did not make a clear case for this sequence specificity in the original manuscript. Based on additional analysis discussed below, we now demonstrate sequence-specific binding by TFAM using AFM, which explains why we observe TFAM-mediated looping primarily for the mtDNA sequence. We also present a hypothesis for how sequence-specific binding in the loop leads to protein-mediated looping at nearby sequences.

*2) Related to the point above: AFM images can visualise proteins bound to DNA directly, a strength of this type of imaging. Most images shown in this work do not indicate the presence of proteins except in rather large complexes, much larger than expected of a TFAM protein. The authors should present quantitative data on TFAM-DNA binding, information that should be easily extracted from the images:% DNA bound by protein, position of proteins along the contour of DNA, size of DNA-bound proteins.*

We thank the reviewers for these excellent suggestions. We have used our AFM images to locate the position and size of each detectable protein cluster on the DNA (Figure 7—figure supplement 6 and Figure 7—figure supplement 10). From this information, we quantitatively characterize the DNA and protein configurations for both mtDNA and the puc18 control construct. Figure 7—figure supplement 7 shows the percent of DNA molecules (either mtDNA or puc18) in each of four categories: Loop present but no protein, loop present with protein bound, no loop with protein bound, and no loop and no protein bound. For mtDNA, we see a strong peak in the category of loop formed with protein bound, while for puc18 these categories are all equally probable. This supports the preferential binding of TFAM to mtDNA and demonstrates how this preference leads to protein-mediated loops at the demonstrated concentration of 10 nM protein added. To fully quantify the binding affinity for these constructs, we also find the fractional binding of TFAM on each DNA lattice by determining the fraction of the DNA contour that is covered by protein as a function of protein concentration. Figure 7—figure supplement 8 shows the measured fractional binding as a function of effective concentration of TFAM in solution (taking into account the reduction in protein concentration due to DNA binding). The results clearly demonstrate a factor of 3 increase in TFAM binding affinity for the mtDNA construct (K_d_=59 ± 2 nM) relative to the puc18 construct (K_d_=200 ± 40 nM) under our solution conditions. Because TFAM is a DNA bending protein, such an affinity increase due to the IPR sequences located in the loop could easily lead to nucleation of protein complexes at the loop crossing point. Thus, sequence-specific binding within the loop leads to protein binding at crossover points, even when these crossover points are not the preferred sequence. This information is now discussed in detail in the manuscript, providing a direct connection between the sequence-specific transcription measurements and AFM measurements of DNA binding.

The reviewer also notes the observation of what seem to be large protein complexes. We have also characterized the size of the protein complexes. We first measured the volume distribution of individual proteins, revealing an average volume of 61 nm^[3]^ (Figure 7—figure supplement 9) and an average height of 0.98 nm (Figure 7—figure supplement 9). We next measured the distribution of protein complex volume when bound to DNA. We find that the average volume of protein at the DNA crossing is 138 ± 15 nm^[3]^, or 2 to 3 proteins. When the complexes are found outside the crossing point, the average volume is 103 ± 9 nm^[3]^, or 1 to 2 proteins (Figure 7—figure supplement 10). These measurements support the hypothesis that crossing points nucleate TFAM cluster formation, consistent with the model presented above. The cluster sizes observed are consistent with previous measurements of the cooperativity of TFAM binding (ω=70, Farge et al. Nature Communications 2012).

*3) The optical tweezers experiment is not well explained in the Materials and methods or Results sections. The concentration of TFAM is not given. Λ DNA is used, so isn't all binding non-specific? It is assumed that rupture events represent protein-protein dimer breakage whereas any number of other possibilities exist (e.g. protein-DNA). Statements about protein disassociation in time are presented without any (quantitative) analysis or reference to known properties. The data as presented do not add usefully to this story – the authors should carefully revise.*

The optical tweezers measurements are indeed only able to observe nonsequence-specific looping. However, we are not arguing that looping does not occur for random sequences, but that looping is enhanced at mtDNA sequences. The optical tweezers experiments provide support for the observation that TFAM stabilizes loops in solution. Such measurements are important to eliminate the possibility that the looping observed in AFM is due to surface interactions. We therefore believe it is important to include the data demonstrating loop formation with optical tweezers. However, we agree with the reviewer that the measurements of protein dissociation in time are not needed and we have removed this discussion.

*4) TFAM is stated to be a dimer based only on a weak interface observed in a crystal. The importance of the C-terminus is clear but the relevance of dimerization is not supported. The authors should provide or cite quantitative date for the multimerization of TFAM in biochemically relevant conditions.*

The quantitative analysis of the AFM data discussed above now demonstrates that the protein clusters found at loop crossing points are on average dimers, providing quantitative data for multimerization of TFAM at 10 nM, a concentration shown to be biochemically relevant in transcription assays (Figure 7—figure supplement 10).

*5) The observation that subtle changes in TFAM concentration have significant effects on the looping/compaction of DNA and the differential transcription from the mitochondrial promoters is not new. Surprisingly, the authors make no mention of the 2014 Cell Reports paper by Farge and co-workers that provide insight that is highly relevant to this work.*

We apologize for this oversight. We were aware of this work but the reference was inadvertently removed during editing. We have added the reference to the manuscript.